# Unravelling the Performance of Physics-informed Graph Neural Networks for Dynamical Systems

**Abishek Thangamuthu, Gunjan Kumar**
Department of Computer Science and Engineering
Indian Institute of Technology Delhi
New Delhi 110016, India

**Suresh Bishnoi**
School of Interdisciplinary Research
Indian Institute of Technology Delhi
New Delhi 110016, India

**Ravinder Bhattoo**
Department of Civil Engineering
Indian Institute of Technology Delhi
New Delhi 110016, India

**N M Anoop Krishnan, Sayan Ranu**
Yardi School of Artificial Intelligence
Indian Institute of Technology Delhi
New Delhi 110016, India
`{krishnan,sayanranu}@iitd.ac.in`

## Abstract

Recently, graph neural networks have been gaining a lot of attention to simulate dynamical systems due to their inductive nature leading to *zero-shot generalizability*. Similarly, physics-informed inductive biases in deep-learning frameworks have been shown to give superior performance in learning the dynamics of physical systems. There is a growing volume of literature that attempts to combine these two approaches. Here, we evaluate the performance of thirteen different graph neural networks, namely, *Hamiltonian* and *Lagrangian* graph neural networks, graph *neural ODE*, and their variants with explicit constraints and different architectures. We briefly explain the theoretical formulation highlighting the similarities and differences in the inductive biases and graph architecture of these systems. We evaluate these models on spring, pendulum, gravitational, and 3D deformable solid systems to compare the performance in terms of rollout error, conserved quantities such as energy and momentum, and generalizability to unseen system sizes. Our study demonstrates that GNNs with additional inductive biases, such as explicit constraints and decoupling of kinetic and potential energies, exhibit significantly enhanced performance. Further, all the physics-informed GNNs exhibit zero-shot generalizability to system sizes an order of magnitude larger than the training system, thus providing a promising route to simulate large-scale realistic systems.

## 1 Introduction and Related Works

Understanding the time evolution or "dynamics" of physical systems is a long-standing problem of interest with both fundamental and practical relevance in areas of physics, engineering, mathematics, and biology [1, 2, 3, 4, 5, 6]. Traditionally, the dynamics of physical systems are expressed in terms of differential equations with respect to their state variables, such as the position $(\mathbf{x}(t))$ or velocity $(\dot{\mathbf{x}}(t))$ [1, 3]. Note that the state variables are actual observable quantities and define the configurational state of a system at any point of time $t$. The differential equation is then solved with the initial and boundary conditions to predict the future states of the system. While this conventional approach is highly efficient and requires little data to predict the dynamics, in many cases the exact differential equation may be unknown. Further, the formulation of the differential equations might require the knowledge of abstract quantities such as the energy, or force distribution of the system, which are not directly measurable in most cases [7].

To this extent, neural networks or MLPs present as efficient function approximators, that can learn the dynamics directly from the state or trajectory [2, 8]. The learned dynamics can then be used to

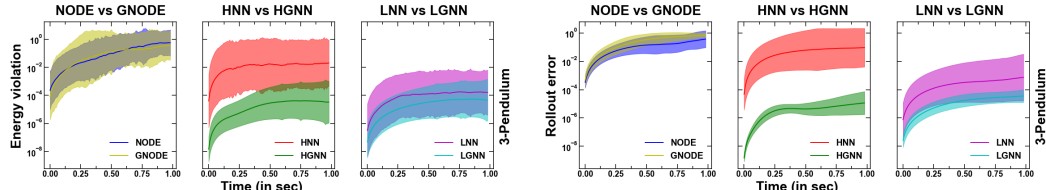

Figure 1: Comparison of physics-informed neural networks (NODE, HNN, and LNN) and their graph-counterparts (GNODE, HGNN, LGNN) for 3-pendulum systems. The error bar represents the 95% confidence interval over 100 trajectories generated from random initial conditions. Further details of the architectures, and the datasets are discussed in § 3 and § 4.1 respectively.

infer the future states of the system with different initial conditions. It has been shown recently that the learning can be significantly enhanced if *physics-based inductive biases* are provided to these MLPs [9, 10, 11, 12, 13]. Specifically, these biases allow the MLP to preserve the characteristics of physical systems, such as energy and momentum conservation, and thus lead to a realistic realization of a trajectory of the system [14, 9, 15]. The most popular choices for learning dynamics are the *neural ODE* (NODE) [8, 16, 13], *Lagrangian* (LNNs) [9, 10, 17, 12], and *Hamiltonian* neural networks (HNNs) [18, 14, 19, 20]. While NODE learns the differential equations as a parameterized neural network [8], LNN and HNN learn the *Lagrangian* and *Hamiltonian* of the system, respectively, which is then used to predict the forward trajectory through the physics-based equation. It is worth noting that in all these cases, the training is purely based on the trajectory or state of the system [21].

One of the major disadvantages of these systems is that they are *transductive* in nature, that is, they work only for the systems they are trained for. For instance, an LNN trained for a double pendulum can be used to infer the dynamics of a double pendulum only and not any $n$-pendulum. This significantly limits the application of LNN, HNN, and NODE to simple systems since for each system the training trajectory needs to be generated and the model needs to be trained. It has been shown the transductivity of MLPs could be addressed by incorporating an additional inductive bias in the structure by accounting for the topology of the system in the graph framework using a graph neural network (GNN) [22, 23, 7, 18, 12, 13]. GNNs, once trained has the capability to generalize to arbitrary system sizes.

It is worth noting that most of the studies on GNNs for dynamical systems use a purely data-driven approach, where the GNNs are used to learn the position and velocity update directly from the data. Recently, physics-informed GNNs have been used to simulate complex physical systems [23]. In addition, since the GNNs are trained at the node and edge level, they can potentially learn more efficiently from the same number of data points in comparison to their fully connected counterparts. Figure 1 shows the energy and rollout error of NODE, LNN, and HNN with their graph-based counterparts, namely, GNODE, LGNN, and HGNN (detailed structure discussed in Section 3) for 3-pendulum systems. We observe that the graph-based versions provide comparable or better performance than their counterparts when trained on the same amount of data for the same number of epochs. While this difference is not significant in NODE, in LGN and HGN, we observe that the difference is more pronounced. In addition, as demonstrated later in this work (see Section 4.3.3), the graph-versions have the additional advantage of inductivity to generalize to larger system sizes.

Despite the wide use of GNNs to model physical systems, there exist no work thus far that systematically benchmark their performance for learning and inferring the dynamics of physical systems. In this work, we aim to address this gap by evaluating the role of different physics-based inductive biases in GNNs for learning the dynamics of complex systems such as $n$-spring and pendulum systems, gravitational bodies, and elastically deformable 3D solid. The major contributions of the present work are as follows.

1. **Topology-aware modeling.** We benchmark thirteen different physics-informed GNNs that model physical systems. By analyzing their energy, rollout, and momentum violation, we carefully evaluate and benchmark their performance.
2. **Physics-informed inductive bias.** We analyze the nature of inductive bias provided for GNNs by Lagrangian, Hamiltonian and Newtonian mechanics in learning the dynamics, although they, in principle, are equivalent formulations.
3. **Explicit constraints.** We analyze the role of providing explicit constraints as an inductive bias on the performance of different GNNs.
4. **Decoupling potential and kinetic energies.** We analyze the effect of exploiting the Hamiltonian and Lagrangian structure, which allows decoupling of kinetic and potential energies in the

Cartesian coordinates. Specifically, we evaluate how this decoupling, by modifying the graph architecture, affect the performance of GNNs.

5. **Zero-shot generalizability.** Finally, we evaluate the ability of these GNNs to generalize to arbitrary system sizes an order of magnitude larger than the systems they have been trained on.

## 2    Preliminaries on Learning Dynamical Systems

We consider a rigid body system comprising of $n$ interacting particles. The configuration of this system is represented as a set of Cartesian coordinates $\mathbf{x}(t) = (x_1(t), x_2(t), \ldots, x_n(t))$. Since we are using a graph neural network to model the physical interactions, it is natural to select the Cartesian coordinates for the features such as position as velocity. While this may result in an increased complexity in the form of the Hamiltonian or Lagrangian, it significantly simplifies the mass matrix for particle-based systems by making it positive definite [10] and diagonal [10, 24].

### 2.1    ODE Formulation of Dynamics

Traditionally, the dynamics of a system can be expressed in terms of the *D'Alembert's principle* as [3]

$$\mathbf{M}\ddot{\mathbf{x}} - \mathbf{F}(\mathbf{x}, \dot{\mathbf{x}}) = 0 \tag{1}$$

where, in Cartesian coordinates, $\mathbf{M}$ is the constant mass matrix that is independent of the coordinates [24], and $\mathbf{F}$ represents the dynamics of the system. Accordingly, the acceleration $\ddot{\mathbf{x}}$ of the system can be computed as:
$$\ddot{\mathbf{x}} = \mathbf{M}^{-1}(\mathbf{F}(\mathbf{x}, \dot{\mathbf{x}})) \tag{2}$$

In case of systems with constraints, for instance a pendulum system where the length between the two bobs are maintained constant, Eq. 1 can be modified to feature the constraints explicitly [3, 4].
$$\mathbf{M}\ddot{\mathbf{x}} - \mathbf{F}(\mathbf{x}) + A^T\lambda = 0 \tag{3}$$

Here, the constraints of the system are given by $A(\mathbf{x})\dot{\mathbf{x}} = 0$, where $A(\mathbf{x}) \in \mathbb{R}^{k \times D}$ represents $k$ constraints associated with the $D$ degrees of freedom (See App. A for details on why constraints are expressed through this form). Thus, $A$ represents the non-normalized basis of constraint forces given by $A = \nabla_{\mathbf{x}}(\phi)$, where $\phi$ is the matrix corresponding to the constraints of the system and $\lambda$ is the *Lagrange multiplier* corresponding to the relative magnitude of the constraint forces.

To solve for $\lambda$, the constraint equation can be differentiated with respect to time to obtain $A\ddot{\mathbf{x}} + \dot{A}\dot{\mathbf{x}} = 0$. Substituting for $\ddot{\mathbf{x}}$ from Eq. 3 and solving for $\lambda$, we get:
$$\lambda = (A\mathbf{M}^{-1}A^T)^{-1}(A\mathbf{M}^{-1}(\mathbf{F}) + \dot{A}\dot{\mathbf{x}}) \tag{4}$$

Accordingly, $\ddot{\mathbf{x}}$ can be obtained as:
$$\ddot{\mathbf{x}} = \mathbf{M}^{-1}\left(\mathbf{F} - A^T(A\mathbf{M}^{-1}A^T)^{-1}\left(A\mathbf{M}^{-1}(\mathbf{F}) + \dot{A}\dot{\mathbf{x}}\right)\right) \tag{5}$$

### 2.2    Lagrangian Dynamics

The standard form of Lagrange's equation for a system with *holonomic* constraints is given by
$$\frac{d}{dt}\left(\nabla_{\dot{\mathbf{x}}}L\right) - \left(\nabla_{\mathbf{x}}L\right) = 0 \tag{6}$$

where the Lagrangian is $L(\mathbf{x}, \dot{\mathbf{x}}, t) = T(\mathbf{x}, \dot{\mathbf{x}}, t) - V(\mathbf{x}, t)$ with $T(\mathbf{x}, \dot{\mathbf{x}}, t)$ and $V(\mathbf{x}, t)$ representing the total kinetic energy of the system and the potential function from which generalized forces can be derived, respectively. Accordingly, the dynamics of the system can be represented using *Euler-Lagrange (EL)* equations as
$$\ddot{\mathbf{x}} = (\nabla_{\dot{\mathbf{x}}\dot{\mathbf{x}}}L)^{-1}\left[\nabla_{\mathbf{x}}L - (\nabla_{\dot{\mathbf{x}}\mathbf{x}}L)\dot{\mathbf{x}}\right] \tag{7}$$

Here, $\nabla_{\dot{\mathbf{x}}\dot{\mathbf{x}}}$ refers to $\frac{\partial^2}{\partial\dot{\mathbf{x}}^2}$. In systems with constraints, the Lagrangian formulation can be modified to include the explicit constraints [3]. Accordingly, the acceleration can be computed as
$$\ddot{\mathbf{x}} = \nabla_{\dot{\mathbf{x}}\dot{\mathbf{x}}}L\left(-\nabla_{\dot{\mathbf{x}}\mathbf{x}}L\dot{\mathbf{x}} - \nabla_{\mathbf{x}}L - A^T(A(\nabla_{\dot{\mathbf{x}}\dot{\mathbf{x}}}L)^{-1}A^T)^{-1}\left(A(\nabla_{\dot{\mathbf{x}}\dot{\mathbf{x}}}L)^{-1}(-\nabla_{\dot{\mathbf{x}}\mathbf{x}}L\dot{\mathbf{x}} - \nabla_{\mathbf{x}}L) + \dot{A}\dot{\mathbf{x}}\right)\right) \tag{8}$$

In Cartesian coordinates, the Lagrangian simplifies to $L(\mathbf{x}, \dot{\mathbf{x}}) = \frac{1}{2}\dot{\mathbf{x}}^T\mathbf{M}\dot{\mathbf{x}} - V(\mathbf{x})$. Exploiting the structure of Lagrangian by decoupling the kinetic and potential energies, and substituting this expression in Eq. 8, we obtain $\mathbf{M} = \nabla_{\dot{\mathbf{x}}\dot{\mathbf{x}}}L$ as a constant matrix independent of coordinates, $\mathbf{C} = \nabla_{\dot{\mathbf{x}}\mathbf{x}}L = 0$, and $\mathbf{F} = \nabla_{\mathbf{x}}V(\mathbf{x})$. Accordingly, the $\ddot{\mathbf{x}}$ can be obtained as
$$\ddot{\mathbf{x}} = \mathbf{M}^{-1}\left(-\nabla_{\mathbf{x}}V(\mathbf{x}) - A^T(A\mathbf{M}^{-1}A^T)^{-1}\left(A\mathbf{M}^{-1}(-\nabla_{\mathbf{x}}V(\mathbf{x})) + \dot{A}\dot{\mathbf{x}}\right)\right) \tag{9}$$

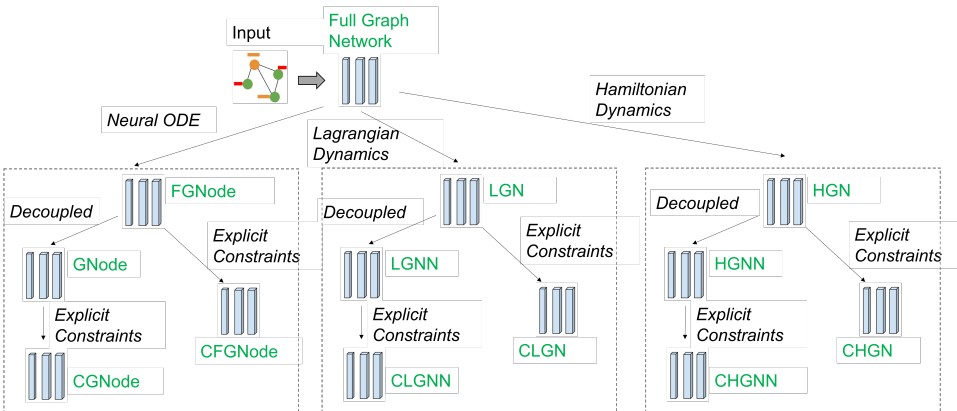

Figure 2: An overview of the various models being benchmarked and the relationship between them. Here, "Decoupled" indicates the model variety where the kinetic and potential energies are decoupled with the force being a function of only the position and the mass matrix being diagonal in nature.

### 2.3 Hamiltonian Dynamics

Hamiltonian equations of motion are given by

$$\mathbf{x} = \nabla_{\mathbf{p_x}} H, \qquad \dot{\mathbf{p}}_{\mathbf{x}} = -\nabla_{\mathbf{x}} H \tag{10}$$

where, $\mathbf{p_x} = \nabla_{\dot{\mathbf{x}}} L = \mathbf{M}\dot{\mathbf{x}}$ represents the momentum of the system in Cartesian coordinates and $H(\mathbf{x}, \mathbf{p_x}) = \dot{\mathbf{x}}^T \mathbf{p_x} - L(\mathbf{x}, \dot{\mathbf{x}}) = T(\dot{\mathbf{x}}) + V(\mathbf{x})$ represents the Hamiltonian of the system. The equation can be simplified by assuming $Z = [\mathbf{x}; \mathbf{p_x}]$ and $J = [0, I; -I, 0]$ then the Hamiltonian equation can be written as

$$\nabla_Z H + J\dot{Z} = 0 \tag{11}$$

Note that this coupled first order differential equations are equivalent to the Lagrangian Eq. 7.

In systems with constraints, the Hamiltonian equations of motion can be modified to feature the constraints explicitly, similar to the Lagrangian and ODE formulations. Hamiltonian equations with explicit constraints can be written as [24]

$$\nabla_Z H + J\dot{Z} + (D_Z \Psi)^T \lambda = 0 \tag{12}$$

where, $(D_Z \Psi)^T \lambda$ represents the effect of constraints on both the equations. Here, $\Psi(Z) = (\Phi; \dot{\Phi})$ and $\Phi = \Phi(\mathbf{x}) = 0$ represent the constraints equation, which implies $\Psi(Z) = 0$. Thus, $(D_Z \Psi)\dot{Z} = 0$. Substituting for $\dot{Z}$ and solving for $\lambda$ yields

$$\lambda = -[(D_Z \Psi)J(D_Z \Psi)^T]^{-1}[(D_Z \Psi)J(\nabla H)] \tag{13}$$

Substituting this value into the Eq. 12 and solving for $\dot{Z}$ yields

$$\dot{Z} = J[\nabla_Z H - (D_Z \Psi)^T [(D_Z \Psi)J(D_Z \Psi)^T]^{-1}(D_Z \Psi)J\nabla_Z H] \tag{14}$$

### 2.4 Physics-informed GNNs

Data-driven GNNs take a given configuration as input and predict the updated configuration [22]. Instead, physics-informed GNNs take the configuration (position and velocity) as input and predict abstract quantities such as *Lagrangian* or *Hamiltonian* or force. These output values are then used along with physics-based equations (as defined in § 2.1-2.3) to obtain the updated trajectory. Thus, in physics-informed GNNs, the neural network essentially learns the function relating the position and velocity to quantities such as force or energy, by training on trajectory.

## 3 Models Studied

In this section, we describe the models studied in this work. An overview of the models and the relationship between them is presented in Fig. 2. Specifically, the full graph network [23] lies at the root, i.e., the base GNN model. On this basic framework, we study the impact of physics-based inductive biases in the form of Neural ODE, Lagrangian and Hamiltonian dynamics. Each of these inductive biases is further refined through the injection of explicit constraints and "decoupled" modeling of the potential and kinetic energies. We next explain each of these specific models. Note

that the detailed architecture of the models are not discussed since they are available in the original papers that introduced the works. Accordingly, we mention the names of the models used in this study, and their major features.

• **Full Graph Network (FGNN)** [23]: FGNN is a *message-passing* GNN, wherein each node $v$ (and potentially edges as well) draws "messages" from its 1-hop neighbors (i.e., their representations in the form of feature vectors), and passes these messages through a multi-layer perceptron (MLP) to construct $v$'s own representation. An FGNN of $\ell$ layers, therefore, learns node representations that capture the topology of the $\ell$-hop neighborhood around each node. FGNN attempts to directly predict the property being modeled (such as acceleration) without exploiting any physics-informed inductive biases.

• **FGNODE**[18]: FGNODE refers to a neural ODE version of FGNN. In FGNODE, the particle positions and velocities are given as the input to each node, while the difference in the position of nodes is given as a feature to each edge. The node-level output of the model is the acceleration on each node, which is then integrated to obtain the updated position and velocity using Eq. 2.

• **GNODE:** GNODE is equivalent to FGNODE with a minor modification in the architecture [13]. Specifically, we note that in particle-based systems in Cartesian coordinates, the kinetic and potential energies can be *decoupled* with the force being a function of only the position and the mass matrix being diagonal in nature [10, 17, 24, 13]. Thus, for GNODE, only the position of the particle is given as a node input and not the velocity, which is then used to predict the node-level acceleration using Eq. 2. Further details of this architecture is provided in App. C.

• **CFGNODE and CGNODE:** CFGNODE and CGNODE are FGNODE and GNODE with explicit constraints [10] as given by Eq. 5, respectively.

• **Lagrangian Graph Network (LGN)** [9]: Here, the graph architecture, which is an FGNN, directly predicts Lagrangian of the system. The acceleration is then predicted from the Lagrangian using *Euler-Lagrange* equation (Eq. 8).

• **LGNN:** LGNN improves on LGN, where the kinetic and potential energies are decoupled [12]. Further, the potential energy is predicted using a GNN with the position as the node input and distance between the nodes as the edge input. The diagonal nature of mass matrix in particle-based rigid body systems in Cartesian coordinates is exploited to learn the parametric masses [9, 10, 17]. Further details of this architecture is provided in App. D.

• **CLGN and CLGNN:** CLGN and CLGNN refers to LNN and LGNN with explicit constraints respectively. Specifically, the acceleration is computed using Eq. 9 [12, 3, 10].

•**HGN:** HGN refers to Hamiltonian graph network, where the Hamiltonian of a system is predicted using the full graph network [23, 18]. Further, the acceleration is computed using Eq. 11.

•**HGNN:** HGNN refers to Hamiltonian graph neural network, where the structure of the Hamiltonian of a system is exploited to decouple it into potential and kinetic energies [12, 18, 14]. Further, the potential energy is predicted using the GNN and the diagonal mass matrix is trained as a learnable parameter. The acceleration is then predicted using Eq. 11. Further details of HGNN is provided in App. E.

• **CHGN and CHGNN:** CHGN and CHGNN refer to HGN and HGNN with explicit constraints respectively (Eq. 12-Eq. 14).

## 4 Benchmarking Evaluation

In this section, we conduct in-depth empirical analysis of the architectures discussed in § 3. To evaluate the models, we consider four systems (see Fig. 3), namely, $n$-pendulums with $n = (3, 4, 5)$, $n$-springs where $n = (3, 4, 5)$, 4-body gravitational system and an elastically deformable 3D solid discretized as particles. All the simulations and training were carried out in the JAX environment [25, 26]. The graph architecture was developed using the jraph package [27]. The experiments were conducted on a linux machine running Ubuntu 18.04 LTS with Intel Xeon processor and 128GB RAM. All codes and data used in the benchmarking study are available at https://github.com/M3RG-IITD/benchmarking_graph and https://doi.org/10.5281/zenodo.7015041, respectively.

### 4.1 Data Generation and Training

**Data generation.** The training data is generated by forward simulation using the known kinetic and potential energies of the systems employing physics-based equations. The timestep used for the forward simulation of the pendulum system is $10^{-5}s$ with the data collected every 1000 timesteps. For

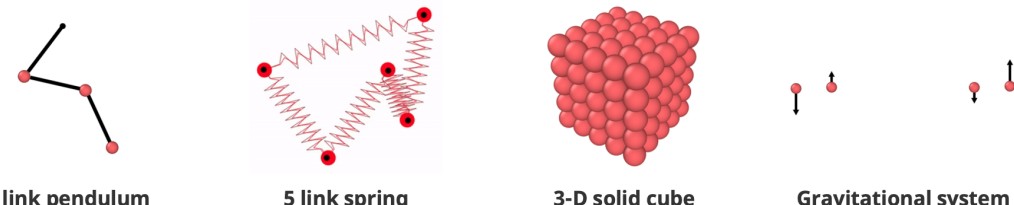

**3 link pendulum**  **5 link spring**  **3-D solid cube**  **Gravitational system**

Figure 3: (From left to right) Visualizations of the systems considered, namely, 3-pendulum, 5-spring, elastically deformable 3D solid, and 4-body gravitational system.

the spring system, the timestep is $10^{-3}s$ with the data collected every 100 timesteps. Integration of the equations of motion is performed using the velocity-Verlet algorithm. In both systems, 100 datapoints are collected from each trajectory and 100 such trajectories based on random initial conditions are used to generate ground truth. This resulted in a total of 10,000 training datapoints for each system. Further details of the experimental systems used for the data generation are provided in the App. B.

**Trajectory prediction and training.** Based on the predicted $\ddot{\mathbf{x}}$, the positions and velocities are predicted using the *velocity Verlet* integration. The loss function is computed by using the predicted and actual accelerations at timesteps $2, 3, \ldots, \mathcal{T}$ in a trajectory $\mathbb{T}$, which is then back-propagated to train the GNNs. Specifically, the loss function is as follows.

$$\mathcal{L} = \frac{1}{n} \left( \sum_{i=1}^{n} \left( \ddot{x}_i^{\mathbb{T},t} - \left( \hat{\ddot{x}}_i^{\mathbb{T},t} \right) \right)^2 \right) \tag{15}$$

Here, $\left( \hat{\ddot{x}}_i^{\mathbb{T},t} \right)$ is the predicted acceleration for the $i^{th}$ particle in trajectory $\mathbb{T}$ at time $t$ and $\ddot{x}_i^{\mathbb{T},t}$ is the true acceleration. $\mathbb{T}$ denotes a trajectory from $\mathfrak{T}$, the set of training trajectories. Note that the accelerations are computed directly from the ground truth trajectory using the Verlet algorithm as:

$$\ddot{x}_i(t) = \frac{1}{(\Delta t)^2} [x_i(t + \Delta t) + x_i(t - \Delta t) - 2x_i(t)] \tag{16}$$

Since the integration of the equations of motion for the predicted trajectory is also performed using the same algorithm as: $x_i(t + \Delta t) = 2x_i(t) - x_i(t - \Delta t) + \ddot{x}_i(\Delta t)^2$, this method is equivalent to training from trajectory (i.e, positions).

We use 10000 data points generated from 100 trajectories to train all the models. This dataset is divided randomly in 75:25 ratio as training and validation set. All models were trained till the decrease in loss saturates to less than 0.001 over 100 epochs. The model performance is evaluated on a test set containing 100 forward trajectories of $1s$ in the case of pendulum and $10s$ in the case of spring. Note that this trajectory is $\approx$ 4-5 orders of magnitude larger than the training trajectories from which the training data has been sampled. The dynamics of $n$-body system is known to be chaotic for $n \geq 2$. Hence, all the results are averaged over trajectories generated from 100 different initial conditions.

The default hyper-parameter values are listed in App. G.4. In addition, we have performed extensive hyper-parameter search to measure their effect on the architectures, the details of which are provided in App. G.5.

### 4.2 Evaluation Metric

Following the work of [10], we evaluate performance by computing the relative error in three different quantities as detailed below. **(1) Rollout Error:** Relative error in the trajectory, known as the *rollout error*, is given by $RE(t) = \frac{||\hat{\mathbf{x}}(t) - \mathbf{x}(t)||_2}{||\hat{\mathbf{x}}(t)||_2 + ||\mathbf{x}(t)||_2}$. **(2) Energy violation:** The error in energy violation is given by $EE(t) = \frac{||\hat{H} - H||_2}{(||\hat{H}||_2 + ||H||_2)}$. **(3) Momentum conservation:** The relative error in momentum conservation is $ME(t) = \frac{||\hat{\mathcal{M}} - \mathcal{M}||_2}{||\hat{\mathcal{M}}||_2 + ||\mathcal{M}||_2}$ Note that all the variables with a hat, for example $\hat{\mathbf{x}}$, represent the predicted values based on the trained model and the variables without hat, $\mathbf{x}$, represent the ground truth. To summarize the performance over a trajectory, following previous works [10], we use the *geometric mean* of relative error of each of the quantities above since the error compounds with time.

### 4.3 Results on Pendulum and Spring systems

For springs and pendulums, the models are trained on 5-spring, and 3-pendulum systems. The trained models are evaluated on unseen system sizes to evaluate their performance on *zero-shot*

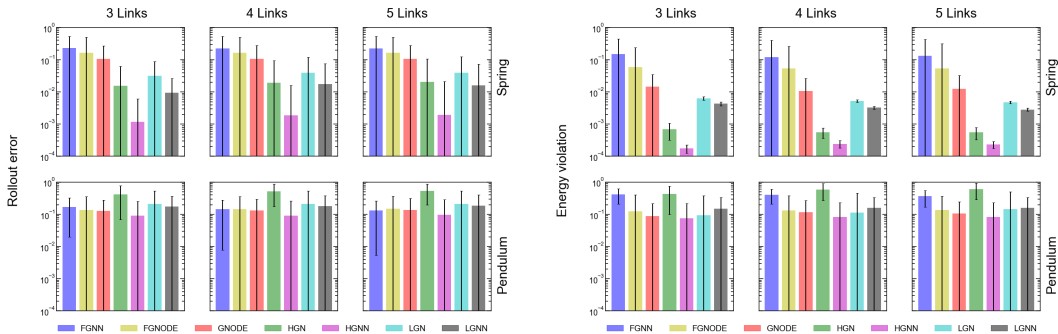

Figure 4: Geometric mean of rollout error and energy error for 3-, 4-, 5-spring and 3-,4-,5-pendulum systems without constraints for LGNN, LGN, HGN, HGNN, GNODE, FGNODE and FGNN. The error bar represents the 95% confidence interval over 100 trajectories generated from random initial conditions.

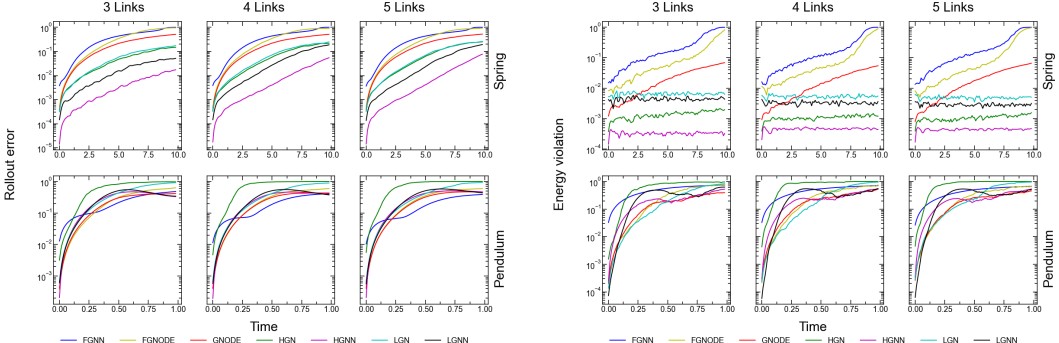

Figure 5: Rollout error and energy error for 3-, 4-, 5-spring and 3-, 4-, 5-pendulum systems without constraints with respect to time for LGNN, LGN, HGN, HGNN, GNODE, FGNODE and FGNN. The curve represents the average over 100 trajectories generated from random initial conditions.

generalizability. To benchmark generalization ability to even larger *unseen* systems, we simulate 5-, 10-, 20-pendulum system and 5-, 20-, 50-spring systems. To compare the performance efficiently, we group the systems without and with explicit constraints separately.

### 4.3.1    Trajectory error and energy conservation

**Systems without explicit constraints.** Figure 4 shows the geometric mean of rollout and energy error of LGNN, LGN, HGN, HGNN, GNODE, FGNODE and FGNN for 3-, 4-, 5-spring and pendulum systems. Fig. 5 shows the evolution of these errors with respect to time. Note that all the systems are trained on 3-pendulum or 5-spring systems alone and performance is evaluated by forward simulation on other systems. The error bar represents the 95% confidence interval obtained based on 100 trajectories generated from different initial conditions. First, we analyze the response on the spring systems (see Fig. 4 top row). We observe that the HGNN exhibits minimum error. This is followed by LGNN and HGN both of which comparable error but higher than that of HGNN. These systems are followed by GNODE. Finally, FGNN exhibits the maximum error. This result is not surprising since FGNN is not physics-informed and learns directly from data.

Now, we focus on the pendulum systems (see Fig. 4 bottom row). Interestingly, in this case, we observe that HGN exhibits a slightly higher error. It is worth noting that the dynamics of spring system is primarily governed by the internal interactions between the balls connected by the springs. In contrast, in pendulum systems, the dynamics is primarily governed by the external gravitational field and the connections simply serve as constraints. These results suggest that in systems, where the dynamics is governed primarily by the internal interactions, the architecture of the graph plays a major role with HGNN exhibiting the best performance for the spring systems. FGNN continues to perform poorly in pendulum systems, particularly in energy violation.

In summary, two key observations emerge from these experiments. First, HGNN consistently provides the lowest error across all setup, with it being more pronounced on spring systems, potentially due to its first-order nature. Second, the decoupled architectures are consistently better (i.e., GNODE and HGNN produces lower error than FGNODE and HGN respectively), which could be attributed to the

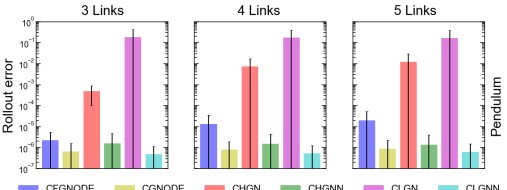
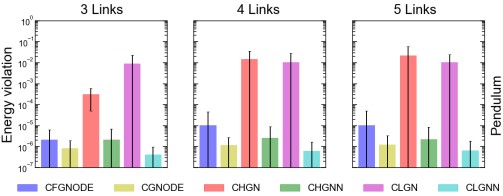

Figure 6: Geometric mean of rollout error and energy error for 3-,4-,5-pendulum systems with constraints for CLGNN, CLGN, CHGN, CHGNN, CGNODE, and CFGNODE. The error bar represents the 95% confidence interval over 100 trajectories generated from random initial conditions.

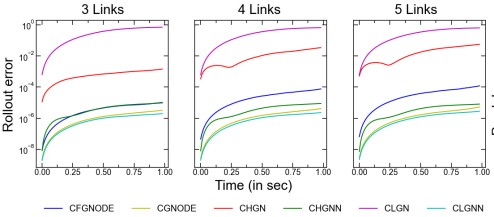
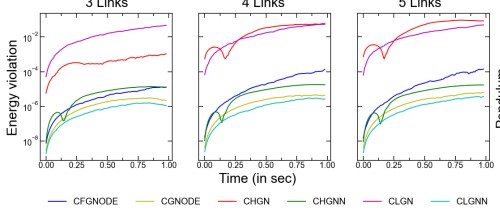

Figure 7: Rollout error and energy error for 3-,4-,5-pendulum systems with constraints with respect to time for CLGNN, CLGN, CHGN, CHGNN, CGNODE, and CFGNODE. The curve represents the average over 100 trajectories generated from random initial conditions.

ability of GNNs to learn the parametric masses and potential energy functions, independently and uniquely. This observation is consistent with previous works on MLPs where decoupling is found to have significantly improved the performance of HNN and LNN [24, 10, 17].

**Systems with explicit constraints.** Figure 6 shows the performance of the constrained architectures for pendulum systems. Figure 7 shows the time evolution of energy and rollout error. Note that in the spring system there are no explicit constraints. Hence, these approaches will be equivalent to learning without constraints in the spring systems as the constraints terms in the equations vanish. We observe that the error in systems with explicit constraints are significantly lower than that without explicit constraints. Similar behavior was observed for fully-connected architectures earlier [10].

We also observe that CLGN exhibits the maximum error on average, followed closely by CHGN. The poor performance of CLGN is not surprising. In pendulum systems, the potential energy of the system depends on the position of the bob. However, in LGN or HGN family of architectures, the actual position of the particle is not given as an input to the graph, rather the edge distance is given as an input. Thus, it is not possible for an LGN or HGN to learn the dynamics of the pendulum system. To address this, we train CLGN and CHGN by giving the position of the bob explicitly as a node input feature. However, despite providing this input, we observe that the final model obtained after training has high loss in comparison to other models.

### 4.3.2 Momentum conservation

While most of the previous studies have focused on energy and rollout error, very few studies have analyzed the momentum error in the trajectory predicted by these systems. It should be noted that conservation of momentum is one of the fundamental laws (as much as the energy conservation), which physical systems are expected to follow. While violation of energy conservation results in spurious generation or dissipation of energy in the system, violation of momentum conservation results in such effects in the

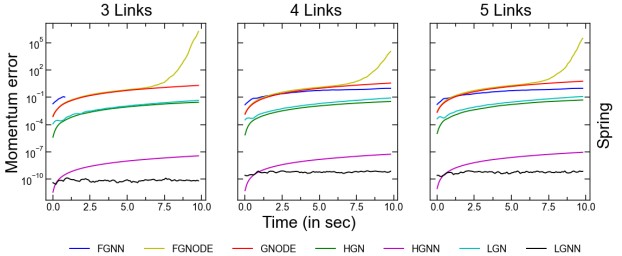

Figure 8: Momentum error for spring systems 3-, 4-, and 5- spring systems for LGNN, LGN, HGN, HGNN, GNODE, FGNODE and FGNN. The curve represents the average over 100 trajectories generated from random initial conditions.

total force on a particle in the system. This, in turn, affects the force equilibrium of a system. Figure 8 shows the evolution of momentum error in FGNN, FGNODE, GNODE, HGN, HGNN, LGN and LGNN for 3-, 4-, and 5-spring systems. Interestingly, we observe that LGNN exhibits least momentum error

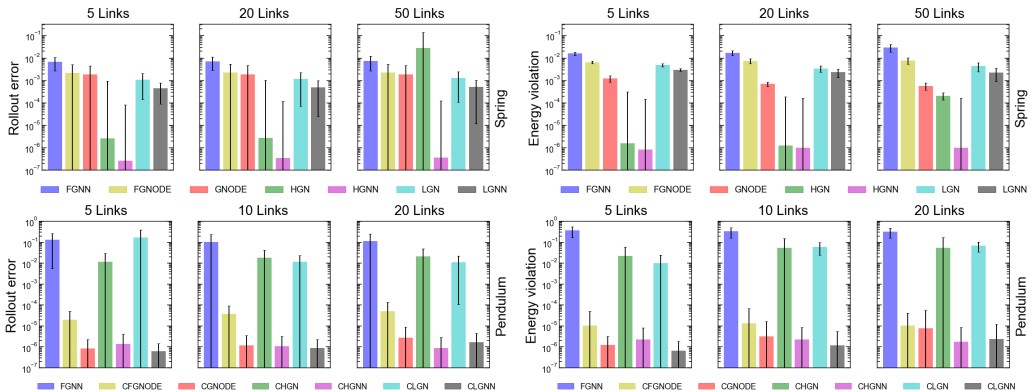

Figure 9: Geometric mean of rollout and energy error for zero-shot generalizability to unseen system sizes for pendulum systems with FGNN, CLGN, CLGNN, CHGN, CHGNN, CGNODE, and CFGNODE and spring systems with FGNN, LGN, LGNN, HGN, HGNN, GNODE, FGNODE. The error bar represents the 95% confidence interval over 100 trajectories generated from random initial conditions.

with a stable value over time. This suggests that the momentum error in the LGNN is saturated and does not diverge over time. Although HGNN exhibits comparable errors, we observe that the value increases with respect to time suggesting a potential divergence at long time scales. This is followed by HGN, which exhibits a momentum error that $\sim 4$ orders of magnitude larger. While FGNODE and GNODE exhibits comparable momentum error at smaller length scales of trajectory, FGNODE diverges significantly faster that GNODE, leading to large unrealistic errors. The improved performed of LGNN and HGNN could be attributed to the decoupling of kinetic and potential energies, and the computation of the potential energy at the edge level, which enforces the momentum conservation indirectly due to the translational symmetry. These results reaffirm that, in interacting systems, the graph architecture and the nature of the inductive bias (NODE vs LNN vs HNN) play crucial roles in governing the stability of the system.

### 4.3.3 Generalizability to unseen system sizes

Now, we analyze the ability of the GNNs to generalize to unseen system sizes. To this extent, we evaluate the performance of the constrained systems for 5-, 10-, and 20-link pendulums and unconstrained systems for 5-, 20-, and 50-link springs. Note that the constrained systems are considered for the pendulum as they exhibit superior performance in comparison to the unconstrained versions. Figure 9 shows the geometric mean of rollout and energy error for these systems. First, we observe that all the graph-based systems exhibit zero-shot generalizability to systems that are almost an order of magnitude larger than the training data with similar errors in both energy and rollout. Further, we observe that the trend in the error for the systems for pendulum and spring systems remain consistent with increasing system sizes. This suggests that there is no additional error added to any of the models considered due to zero-shot generalization. This confirms the superiority of physics-informed GNNs over the traditional MLPs to generate large-scale realistic systems after learning on a significantly smaller system.

### 4.4 Gravitational and 3D solid systems

To test the performance of the models on more complex systems, we consider a 4-body gravitational system and a 3D solid system (see Fig. 3). For gravitational system, a stable four body configuration interacting with each other through the gravitational law and having initial velocities such that they rotate with respect to a common centre is considered (see: App. B). Figure 10 shows the time evolution of energy and rollout error for these systems. We observe that LGNN exhibits the lowest error followed by HGNN. This is followed by LGN and GNODE which exhibits comparable errors.

Now, we evaluate the performance on the elastic deformation of a 3D solid cube. For this, a $5 \times 5 \times 5$ solid cube, discretized into 125 particles, is used for simulating the ground truth in the peridynamics framework. The system is compressed isotropically and then released to simulate the dynamics of the solid, that is, the contraction and expansion in 3D as a function of time. Figure 10 shows the rollout and momentum error for this system (see Fig. 16 for time evolution). Interestingly, we find that the rollout and momentum error for all the models are comparable. This could be attributed to the nature of deformation of the structure, which exhibits only small displacements within the elastic regime. Note that this dynamics is much simpler and non-chaotic in nature. Thus, all the models are able to

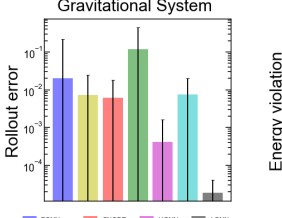 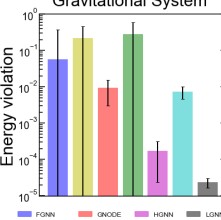 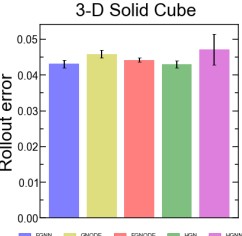 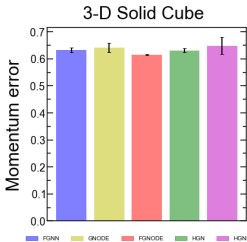

Figure 10: (left to right) Geometric mean of rollout error, energy error for 4-body gravitational system for Lgnn, Lgn Hgn, Hgnn, Gnode, FGnode and Fgnn. Geometric mean of rollout error and momentum error for rigid body system for Hgn, Hgnn, Gnode, FGnode and Fgnn. The error bar represents the 95% confidence interval over 100 trajectories generated from random initial conditions.

learn and infer the dynamics with similar accuracy. It may be concluded from this experiment that for dynamics that is not chaotic and relatively small in terms of magnitude, physics-informed Gnns may not have any additional advantage over the data-driven ones.

## 5 Concluding Insights

In this work, we benchmark the performance of thirteen different physics-informed Gnns for simulating interacting physical systems such as springs and pendulums. The key insights drawn from our study are summarized below:

- **Architecture matters:** We show the nature of the inductive bias provided by the ODE, Hamiltonian and Lagrangian equations lead to different performances for the same graph architecture, although these equations are, in-principle, equivalent [24]. Specifically, the ODE and Lagrangian formulations result in a second-order differential equation, while the Hamiltonian formulation results in a first-order differential equation.
- **Physics-informed Gnns are better:** All of the physics-informed Gnns exhibit better performance than Fgnn, that is learning purely from data except in the case of small displacements that are non-chaotic in nature (elastic deformation of 3D solid). The inductive bias provided for learning and inference by Lagrangian, Hamiltonian, and Newtonian mechanics to Gnns are clearly distinct, although the formulations are equivalent in principle.
- **Constraints help:** Incorporating constraints explicitly as an inductive bias significantly simplifies the learning (see Fig. 14) and enhances the performance of all the graph-based models.
- **Kinetic and potential energies should be decoupled:** We observe that exploiting the fact that the kinetic and potential energies can be decoupled (in Cartesian coordinates) leads to improved performance in all the models—Gnode better than FGnode, Hgnn better than Hgn, and Lgnn better than Lgn for most systems. This decoupling exhibits a significantly stronger enforcement of the momentum conservation error. This could be attributed to the fact that, when the potential and kinetic energies are decoupled, additional spurious cross-terms involving both $\mathbf{x}$ and $\dot{\mathbf{x}}$ may not be learned by the model, which leads to lower error in the governing equation and the momentum error.
- **Zero-shot generalizability:** We show that all the graph-based models generalize to larger systems with comparable error as the smaller systems. Thus, a graph-based simulator with low error in small systems shall also exhibit similar error in significantly larger systems—thanks to their inductive bias.
- **The Lgn family is slow to train:** Due to the double derivative present in the Lagrangian, the Lgn family exhibits highest training and inference times in comparison to other models (See App. H). These slow running times do not provide any benefit on accuracy. Hence, Gnode and Hgn provide better efficiency-efficacy balance while making a Gnn physics-informed.

**Limitations and outlook.** Our results clearly highlight the importance of constraints. Current physics-informed Gnns require these constraints to be explicitly incorporated. An important future direction of research would therefore be to *learn* the constraints directly from the trajectory. In addition, incorporating deformations, contacts, and other realistic phenomena remains to be explored in physics-informed graph architecture, although similar approaches have been employed in fully connected MLPs [19, 28]. Finally, we observe that there is a 10-fold increase in error in the presence of noise in the training data (see App. F.2). This behavior points towards the need to build architectures that are more robust to noise.

## Acknowledgments

The authors thank IIT Delhi HPC facility for computational and storage resources.

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
