# Supplementary: Unravelling the Performance of Physics-informed Graph Neural Networks for Dynamical Systems

**Abishek Thangamuthu, Gunjan Kumar**
Department of Computer Science and Engineering
Indian Institute of Technology Delhi
New Delhi 110016, India

**Suresh Bishnoi**
School of Interdisciplinary Research
Indian Institute of Technology Delhi
New Delhi 110016, India

**Ravinder Bhattoo**
Department of Civil Engineering
Indian Institute of Technology Delhi
New Delhi 110016, India

**N M Anoop Krishnan, Sayan Ranu**
Yardi School of Artificial Intelligence
Indian Institute of Technology Delhi
New Delhi 110016, India
{krishnan,sayanranu}@iitd.ac.in

## A    Expressing constraints

A constraint on a system essentially restricts the motion of a system to a subspace among all the allowable paths. For instance, in the case of two particles with the coordinates $(x, y)$ and $(0, 0)$ connected by an in-extensible rod, the constraint equation can be given as $(x^2 + y^2) = l^2$. Such constraints are known *holonomic* constraints. However, a different set of constraints act in cases such as multi-fingered grasping, known as *Pfaffian* constraints, where instead of positions, constraints are enforced on velocities. The generic form of a Pfaffian constraint is $A(\mathbf{x})\dot{\mathbf{x}} = 0$. Note that any holonomic constraint can also be written in the form of a Pfaffian constraint by differentiating the original form. For instance, the constraint equation for two particles mentioned earlier can be differentiated to obtain $\mathbf{x}\dot{\mathbf{x}} + \mathbf{y}\dot{\mathbf{y}} = 0$, which is of the form $A(\mathbf{x})\dot{\mathbf{x}} = 0$. For the sake of generality, here we adopt this form to express the constraints. More details on this can be found in the Section 1, Chapter 6 of Murray et al. (Murray, R.M., Li, Z. and Sastry, S.S., 2017. A mathematical introduction to robotic manipulation. CRC press.)

## B    Experimental systems

To simulate the ground truth, physics-based equations derived using Lagrangian mechanics is employed. The equations for $n$-pendulum and spring systems are given in detail below.

### B.1    $n$-Pendulum

For an $n$-pendulum system, $n$-point masses, representing the bobs, are connected by rigid (non-deformable) bars. These bars, thus, impose a distance constraint between two point masses as

$$||x_i - x_{i-1}||^2 = l_i^2 \tag{17}$$

where, $l_i$ represents the length of the bar connecting the $(i-1)^{th}$ and $i^{th}$ mass. This constraint can be differentiated to write in the form of a *Pfaffian* constraint as

$$(x_i - x_{i-1})(\dot{x}_i - \dot{x}_{i-1}) = 0 \tag{18}$$

Note that such constraint can be obtained for each of the $n$ masses considered to obtain the $A(q)$.

36th Conference on Neural Information Processing Systems (NeurIPS 2022) Datasets and Benchmarks Track.

The Lagrangian of this system can be written as

$$L = \sum_{i=1}^{n} \left( 1/2 m_i \dot{x_i}^{\mathsf{T}} \dot{x_i} - m_i g x_i^{(2)} \right) \tag{19}$$

where $m_i$ represents the mass of the $i^{th}$ particle, $g$ represents the acceleration due to gravity in the $x^{(2)}$ direction and $x^{(2)}$ represents the position of the particle in the $x^{(2)}$ direction.

## B.2    $n$-spring system

Here, $n$-point masses are connected by elastic springs that deform linearly (elastically) with extension or compression. Note that similar to the pendulum setup, each mass $m_i$ is connected to two masses $m_{i-1}$ and $m_{i+1}$ through springs so that all the masses form a closed connection. The Lagrangian of this system is given by

$$L = \sum_{i=1}^{n} 1/2 m_i \dot{x_i}^{\mathsf{T}} \dot{x_i} - \sum_{i=1}^{n} 1/2 k (||x_{i-1} - x_i|| - r_0)^2 \tag{20}$$

where $r_0$ and $k$ represent the undeformed length and the stiffness, respectively, of the spring.

## B.3    $n$-body gravitational system

Here, $n$ point masses are in a gravitational field generated by the point masses themselves. The Lagrangian of this system is given by

$$L = \sum_{i=1}^{n} 1/2 m_i \dot{x_i}^{\mathsf{T}} \dot{x_i} + \sum_{i=1}^{n} \sum_{j=1, j\neq i}^{n} G m_i m_j / 2 (||x_i - x_j||) \tag{21}$$

where $G$ represents the Gravitational constant.

## B.4    Rigid-body system

Here, in a solid cube of $5 \times 5 \times 5$ size, the dynamics of an elastically deformable body is simulated. Specifically, a $5 \times 5 \times 5$ solid cube, discretized into 125 particles, is used for simulating the ground truth. 3D solid system is simulated using the peridynamics framework. The system is compressed isotropically and then released to simulate the dynamics of the solid, that is, the contraction and expansion in 3D.

## C    Graph Neural ODE (GNODE)

To learn the dynamical systems, GNODEs parameterize the dynamics $F(\mathbf{x}, \dot{\mathbf{x}}, t)$ using a neural network to learn the approximate function $\hat{F}(\mathbf{x}_t, \dot{\mathbf{x}}_t, t)$ by minimizing the loss between the predicted and actual trajectories, that is, $\mathcal{L} = ||\mathbf{x}_{t+1} - \hat{\mathbf{x}}_{t+1}||$. Thus, a GNODE essentially uses graph topology to learn the approximate dynamics $\hat{F}$ by training directly from the trajectory. Figure 11 shows the architecture of the GNODE, which is discussed in detail below.

**Graph structure.** First, an $n$-particle system is represented as a undirected graph $\mathcal{G} = \{\mathcal{V}, \mathcal{E}\}$, where the nodes represent the particles and the edges represents the connections or interactions between them. For example, in pendulum or spring systems, the nodes correspond to bobs or balls, respectively, and the edges correspond to the bars or springs, respectively.

**Input features.** Each node is characterized by the features of the particles, namely, the particle *type* ($t$), *position* ($x_i$), and *velocity* ($\dot{x}_i$). The *type* distinguishes particles of differing characteristics, for instance, balls or bobs with different masses. Further, each edge is represented by the edge features $w_{ij} = (x_i - x_j)$, which represents the relative displacement of the nodes connected by the given edge.

**Pre-Processing.** In the pre-processing layer, we construct a dense vector representation for each node $v_i$ and edge $e_{ij}$ using $\texttt{MLP}_{em}$ as:

$$\mathbf{h}_i^0 = \texttt{squareplus}(\texttt{MLP}_{em}(\texttt{one-hot}(t_i), x_i, \dot{x}_i)) \tag{22}$$

$$\mathbf{h}_{ij}^0 = \texttt{squareplus}(\texttt{MLP}_{em}(w_{ij})) \tag{23}$$

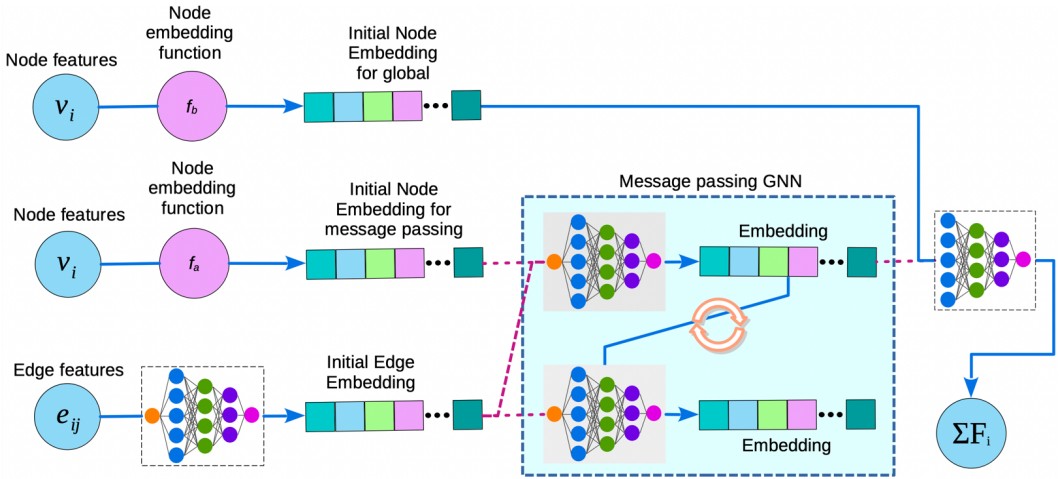

Figure 11: GNODE architecture.

squareplus is an activation function. Note that the $\mathtt{MLP}_{em}$ corresponding to the node and edge embedding functions are parameterized with different weights. Here, for the sake of brevity, we simply mention them as $\mathtt{MLP}_{em}$.

**Acceleration prediction.** In many cases, internal forces in a system that govern the dynamics are closely dependent on the topology of the structure. To capture this information, we employ multiple layers of *message-passing* between the nodes and edges. In the $l^{th}$ layer of message passing, the node embedding is updated as:

$$\mathbf{h}_i^{l+1} = \mathtt{squareplus}\left(\mathbf{h}_i^l + \sum_{j\in\mathcal{N}_i}\mathbf{W}_{\mathcal{V}}^l \cdot \left(\mathbf{h}_j^l||\mathbf{h}_{ij}^l\right)\right) \tag{24}$$

where, $\mathcal{N}_i = \{v_j \in \mathcal{V} \mid e_{ij} \in \mathcal{E}\}$ are the neighbors of $v_i$. $\mathbf{W}_{\mathcal{V}}^l$ is a layer-specific learnable weight matrix. $\mathbf{h}_{ij}^l$ represents the embedding of incoming edge $e_{ij}$ on $v_i$ in the $l^{th}$ layer, which is computed as follows.

$$\mathbf{h}_{ij}^{l+1} = \mathtt{squareplus}\left(\mathbf{h}_{ij}^l + \mathbf{W}_{\mathcal{E}}^l \cdot \left(\mathbf{h}_i^l||\mathbf{h}_j^l\right)\right) \tag{25}$$

Similar to $\mathbf{W}_{\mathcal{V}}^l$, $\mathbf{W}_{\mathcal{E}}^l$ is a layer-specific learnable weight matrix specific to the edge set. The message passing is performed over $L$ layers, where $L$ is a hyper-parameter. The final node and edge representations in the $L^{th}$ layer are denoted as $\mathbf{h}_i^L$ and $\mathbf{h}_{ij}^L$ respectively.

In addition to the internal forces, there could be forces that are independent of the topology and depend only on the features of the particle, for example, gravitational force. To account for these, an additional node embedding that is not included in the message passing, namely, $\mathbf{h}_i^g$ is concatenated with the final node representation after message passing as $\mathbf{z}_i = (\mathbf{h}_i^L||\mathbf{h}_i^g)$. Finally, the acceleration of the particle $\ddot{q}_i$ is predicted as:

$$\ddot{x}_i = \mathtt{squareplus}(\mathtt{MLP}_{\mathcal{V}}(\mathbf{z}_i)) \tag{26}$$

Note that the major difference between GNODE and FGNODE, in addition to the other parametric and architectural differences, is the inclusion of this additional global feature embedding in GNODE, which is absent in FGNODE. As seen earlier, inclusion of this additional embedding significantly improves the performance in cases where there are forces due to external fields such as gravity.

## D  Lagrangian Graph Neural Network (LGNN)

Figure 12 shows the the neural network architecture of the LGNN. The architecture directly predicts the Lagrangian of the system exploiting the topology of the system encoded as graph. Note that the graph structure, input features, pre-processing and the message passing leading to intermediate embedding of nodes and edges for LGNN are developed exactly following the same as GNODE,

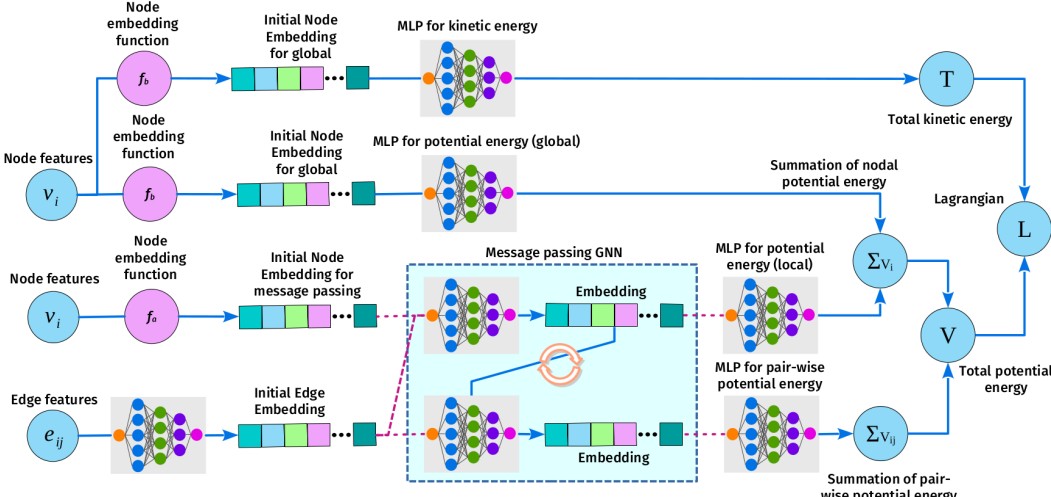

Figure 12: LGNN architecture.

detailed as per section C. Two major differences for LGNN from GNODE is in the computation of the kinetic energy and the potential energy, which are absent in GNODE, as detailed below.

**Kinetic energy.** Since the system is comprised of $n$-point particles, the mass matrix becomes diagonal in Cartesian coordinates [10]. Thus, the kinetic energy, $\tau_i$, of a point particle depends only on its velocity and mass. Here, we learn the mass of each particle based on the node embedding $\mathbf{h}_i^0$ as

$$\tau_i = \texttt{squareplus}(\texttt{MLP}_\tau(\mathbf{h}_i^0 \parallel \dot{x}_i)) \tag{27}$$

The kinetic energy of the individual particles are summed up to compute the total kinetic energy as

$$T = \sum_{u_i \in \mathcal{U}} \tau_i \tag{28}$$

**Potential energy.** Potential energy of a system can have complex combination of absolute and topological features. For instance, a system such as a pendulum in a gravitational field have a simple potential energy function that does not depend on topology. On the contrary, for a system such as balls connected with spring, the potential energy depends on the connections and hence, the topology. Therefore, similar to GNODE, we use final node ($\mathbf{h}_i^L$) and edge ($\mathbf{h}_{ij}^L$) embedding from message passing (representing topology) and global node ($\mathbf{h}_i^g$) embedding to calculate the potential energy of the system as

$$V = \sum_{u_i \in \mathcal{U}} v_i + \sum_{e_{ij} \in \mathcal{E}} v_{ij} \tag{29}$$

where $v_i = \texttt{squareplus}(\texttt{MLP}_{v_i}(\mathbf{h}_i^g) + \texttt{squareplus}(\texttt{MLP}_{\texttt{mp},v_i}(\mathbf{h}_i^L))$ represents the energy due to the attributes of the particle themselves, and $v_{ij} = \texttt{squareplus}(\texttt{MLP}_{v_{ij}}(\mathbf{h}_{ij}^L))$ represents energy due to interactions (topology).

**Lagrangian.** Finally, the Lagrangian of the system is defined as $L = T - V$ where $T$ is total kinetic energy of system and $V$ is the total potential energy of the system. Finally, the acceleration is computed using the predicted Lagrangian employing the appropriate $EL$ equation.

## E  Hamiltonian Graph Neural Network (HGNN)

Figure 13 shows the architecture of the HGNN. Note that HGNN has exactly the same architecture as LGNN and follows all the computations exactly in the same fashion until kinetic and potential energies. Once these energies are obtained, instead of computing the Lagrangian, the Hamiltonian of the system is computed using the equation $H = T + V$ where $T$ is total kinetic energy of system and $V$ is the total potential energy of the system. Finally, the acceleration is computed using the predicted Hamiltonian employing the appropriate Hamiltonian's equation of motion.

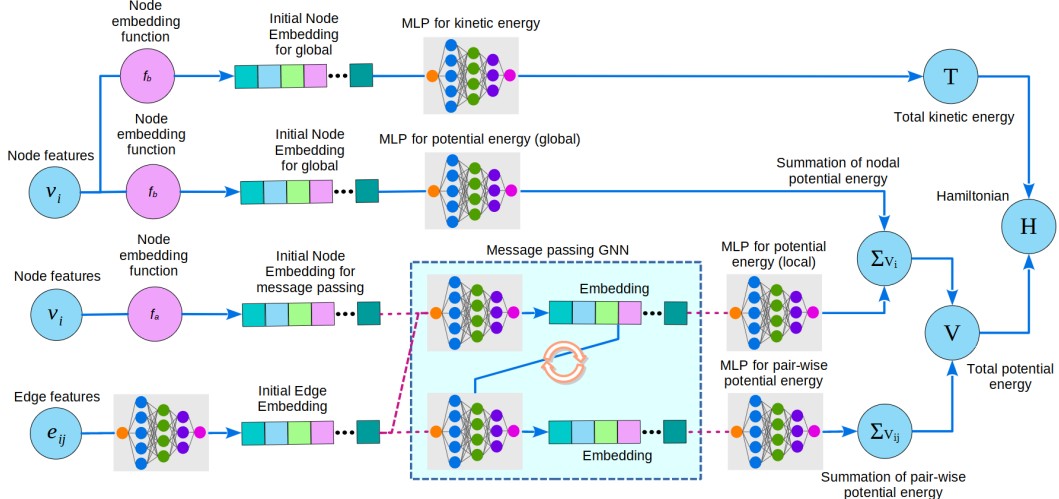

Figure 13: HGNN architecture.

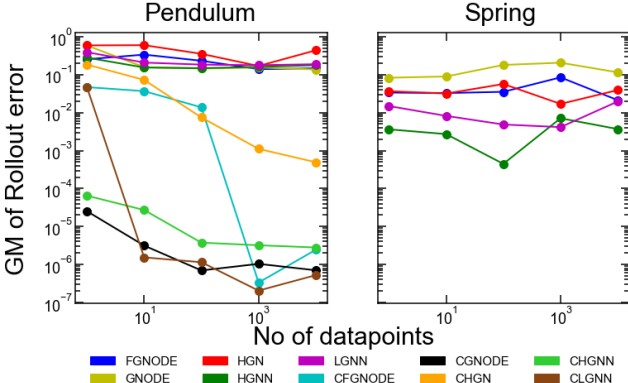

Figure 14: Variation of rollout error against the number of data points used to train the models.

# F Additional Experiments

## F.1 Data Efficiency

Figure 14, shows the dataefficiency of the different models considers. Specifically, we evaluate the rollout error with respect to the number of data-points used to train each of the models. A clear trend emerges from this experiment. Specifically, in pendulum systems, we observe that models with explicit constraints significantly outperform their unconstrained counterparts with more data points by ∼5-6 orders of magnitude. In contrast, the unconstrained architectures show limited reduction in error in both spring and pendulum systems. This trend indicates that injecting explicit constraints in the model leads to more effective training. Further, we observe that the performance of CLGNN, CHGNN, and CGNODE are comparable for pendulum systems, while that of CHGN is poorer, despite having explicit constraints.

## F.2 Robustness to Noise

To evaluate robustness of the evaluated GNNs to noise, we inject Gaussian noise to every data point in the training dataset with mean 0 and standard deviation 1. The forward simulation error is calculated by comparing it to the ground truth trajectories, without adding any noise to those data points. Due to space limitations, the plots are in the appendix. Figure 17 shows the performance on unconstrained architectures, while Figure 18 shows the time evolution of energy and rollout error. Figure 19 and Figure 20 analyze the same, respectively, on pendulum for constrained systems. Finally, we also show the variation of energy and rollout error for spring system for varying percentages of noise, namely, 1%, 5%, 10% and 50% of the standard deviation of the data (see Figs. 21, 22.)

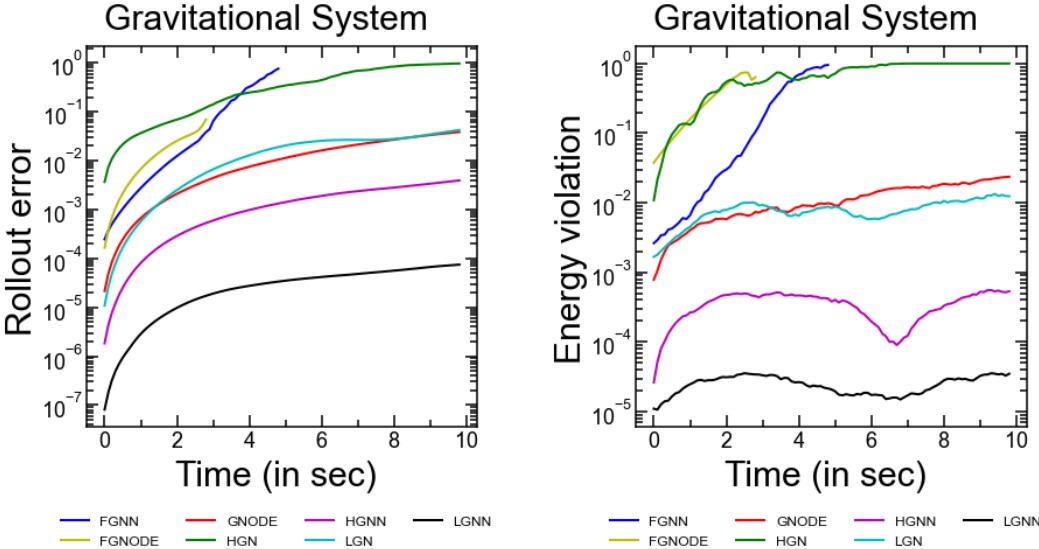

Figure 15: Rollout error and energy error for 4-body gravitational system with constraints with respect to time for LGNN, LGN HGN, HGNN, GNODE, FGNODE and FGNN. The curve represents the average over 100 trajectories generated from random initial conditions.

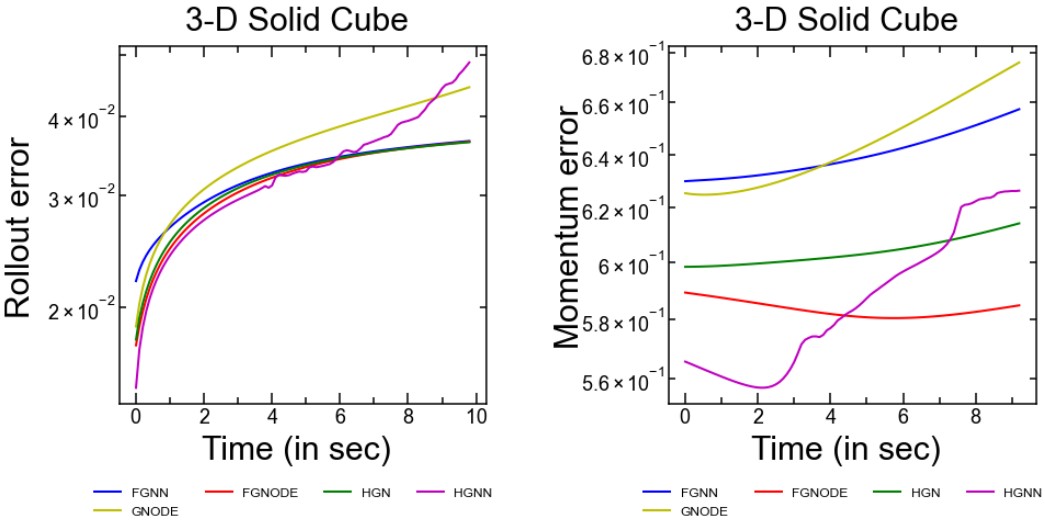

Figure 16: Rollout error and energy error for rigid body system with respect to time for HGN, HGNN, GNODE, FGNODE and FGNN. The curve represents the average over 10 trajectories over the test dataset.

When compared to training on clean data, we observe that the trends remain similar. Specifically, CLGN and CHGN continue to be the poorest performers in the constrained setting. In the unconstrained setting, the same trend continues; FGNN and LGN continue exhibiting highest errors, while HGN remains the best architecture. However, across architectures we observe almost a 10-fold increase in error. All in all, this experiment reveals that while the choice of architectures remain unaffected, all display reduced accuracy. Hence, enabling better robustness would be an important research direction to pursue.

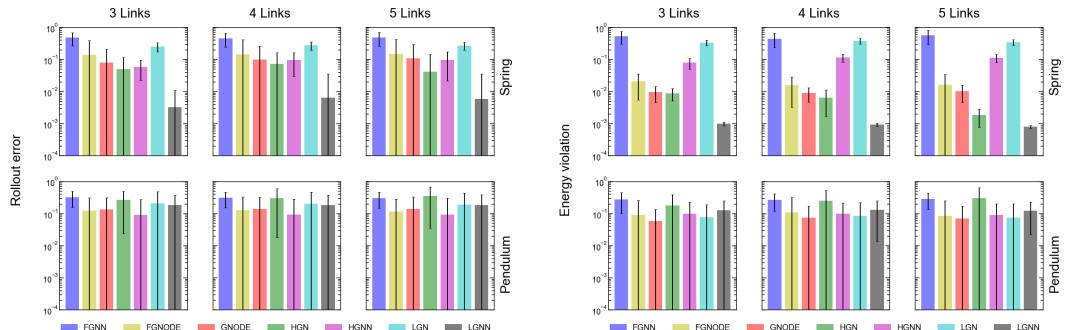

Figure 17: Geometric mean of rollout error and energy error for 3-, 4-, 5-spring and 3-,4-,5-pendulum systems without constraints for LGNN, LGN HGN, HGNN, GNODE, FGNODE and FGNN on noisy data. The error bar represents the 95% confidence interval over 100 trajectories generated from random initial conditions.

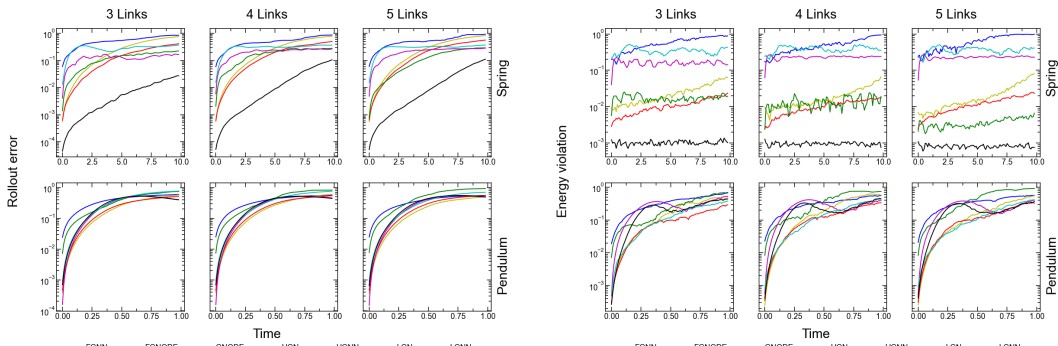

Figure 18: Rollout error and energy error for 3-, 4-, 5-spring and 3-, 4-, 5-pendulum systems without constraints with respect to time for LGNN, LGN HGN, HGNN, GNODE, FGNODE and FGNN on noisy data. The curve represents the average over 100 trajectories generated from random initial conditions.

## G Details of Experimental Setup

### G.1 Dataset generation

**Software packages:** numpy-1.20.3, jax-0.2.24, jax-md-0.1.20, jaxlib-0.1.73, jraph-0.0.1.dev
**Hardware:** Chip: Intel Xeon, Total Number of Cores: 64, Memory: 128 GB, System OS: Ubuntu 18.04.5 LTS.

For all variants of LGNN, LGN, GNODE, FGNODE, FGNN: All the datasets are generated using the known Lagrangian of the pendulum and spring systems, along with the constraints, as described in Section 4. For each system, we create the training data by performing forward simulations with 100 random initial conditions. For the pendulum system, a timestep of $10^{-5}s$ is used to integrate the equations of motion, while for the spring system, a timestep of $10^{-3}s$ is used. The velocity-Verlet algorithm is used to integrate the equations of motion due to its ability to conserve the energy in long trajectory integration.

While for HGN and HGNN, datasets are generated using Hamiltonian mechanics. Runge-Kutta integrator is used to integrate the equations of motion due to the first order nature of the Hamiltonian equations in contrast to the second order nature of LGNN and GNODE.

From the 100 simulations for pendulum and spring system, obtained from the rollout starting from 100 random initial conditions, 100 data points are extracted per simulation, resulting in a total of 10000 data points. Data points were collected every 1000 and 100 timesteps for the pendulum and spring systems, respectively. Thus, each training trajectory of the spring and pendulum systems are

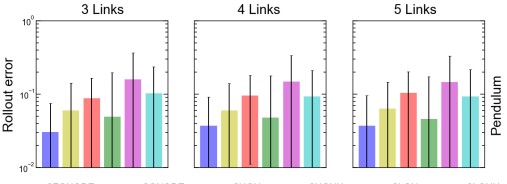
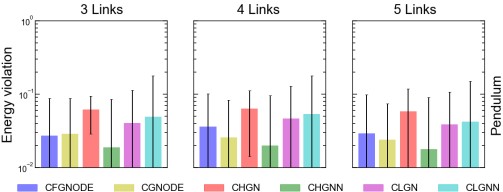

Figure 19: Geometric mean of rollout error and energy error for 3-,4-,5-pendulum systems with constraints for CLGNN, CLGN CHGN, CHGNN, CGNODE, and CFGNODE on noisy data. The error bar represents the 95% confidence interval over 100 trajectories generated from random initial conditions.

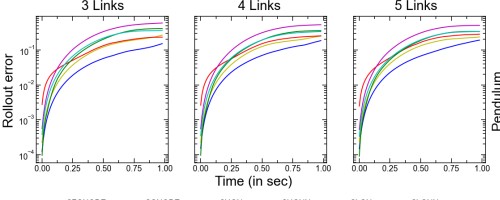
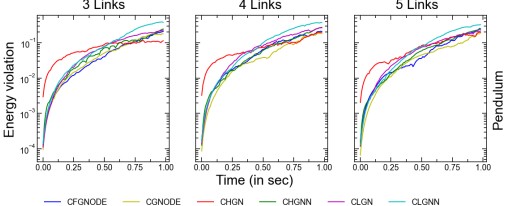

Figure 20: Rollout error and energy error for 3-,4-,5-pendulum systems with constraints with respect to time for CLGNN, CLGN CHGN, CHGNN, CGNODE, and CFGNODE on noisy data. The curve represents the average over 100 trajectories generated from random initial conditions.

$10s$ and $1s$ long, respectively. Here, we do not train from the trajectory. Rather, we randomly sample different states from the training set to predict the acceleration.

For the gravitational system, the dataset is generated using the known Lagrangian of the gravitational system. We create the training data by performing forward simulation from a known stable state. We use a timestep of $10^{-3}s$, which is used to integrate the equations of motion, and similar to the pendulum and spring system, we use the velocity-verlet algorithm for the integration of the equations.

The known stable state is simulated forward and datapoints were collected every 100 timesteps, for a total of 10000 datapoints. Similar to the pendulum and spring systems, we randomly sample different states from the training set to predict the acceleration.

For 3D solid cube, we have generated ground truth data using peridynamics simulation, single initially compressed cube was relaxed for $100s$ with time step $0.1s$. Using this we have generated total 1000 data points.

## G.2 Training details

The training dataset is divided in 75:25 ratio randomly, where the 75% is used for training and 25% is used as the validation set. Further, the trained models are tested on its ability to predict the correct trajectory, a task it was not trained on. Specifically, the pendulum systems are tested for $1s$, that is $10^5$ timesteps, and spring systems for $20s$, that is $2 \times 10^4$ timesteps on 100 different trajectories created from random initial conditions. All models are trained for 10000 epochs. A learning rate of $10^{-3}$ was used with the Adam optimizer for the training.

## G.3 Loss function

Based on the predicted $\ddot{\mathbf{x}}$, the positions and velocities are predicted using the *Velocity Verlet* integration. The loss function is computed by using the predicted and actual accelerations at timesteps $2, 3, \ldots, \mathcal{T}$ in a trajectory $\mathbb{T}$, which is then back-propagated to train the MLPs. Specifically, the loss function is as follows.

$$\mathcal{L} = \frac{1}{n}\left(\sum_{i=1}^{n}\left(\ddot{x}_i^{\mathbb{T},t} - \left(\hat{\ddot{x}}_i^{\mathbb{T},t}\right)\right)^2\right) \tag{30}$$

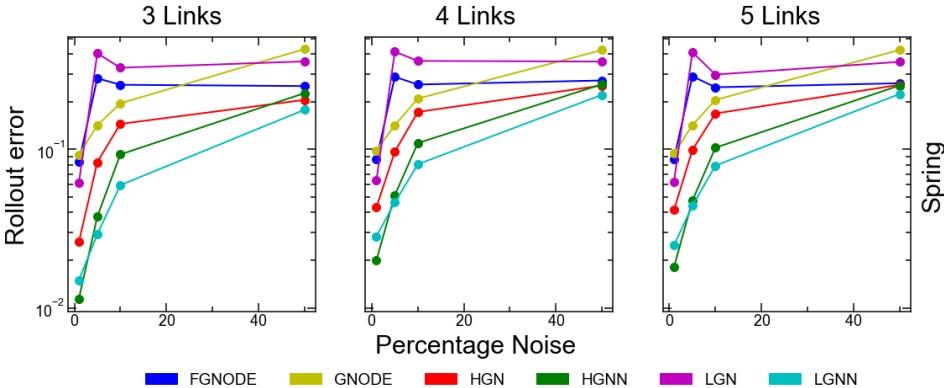

Figure 21: Rollout error with respect to percentage noise in the dataset used to train the 5-spring system.

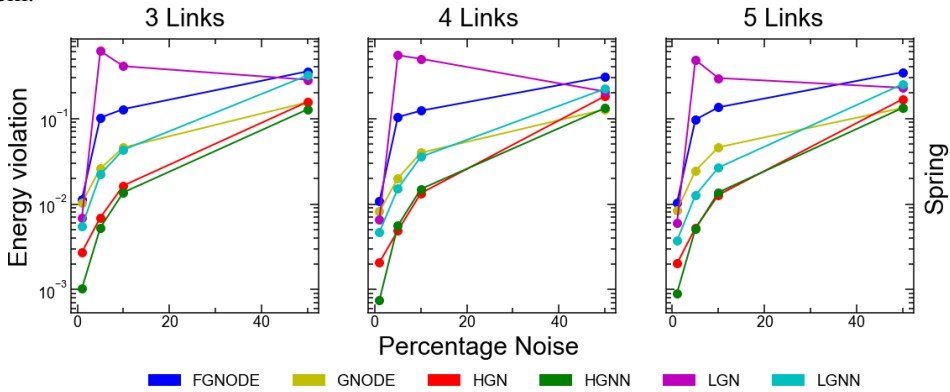

Figure 22: Energy error with respect to percentage of noise in the dataset used to train the 5-spring model.

Here, $(\hat{\ddot{x}}_i^{\mathbb{T},t})$ is the predicted acceleration for the $i^{th}$ particle in trajectory $\mathbb{T}$ at time $t$ and $\ddot{x}_i^{\mathbb{T},t}$ is the true acceleration. $\mathbb{T}$ denotes a trajectory from $\mathfrak{T}$, the set of training trajectories. Note that the accelerations are computed directly from the ground truth trajectory using the Verlet algorithm as:

$$\ddot{\mathbf{x}}(t) = \frac{1}{(\Delta t)^2}[\mathbf{x}(t+\Delta t) + \mathbf{x}(t-\Delta t) - 2\mathbf{x}(t)] \tag{31}$$

Since the integration of the equations of motion for the predicted trajectory is also performed using the same algorithm as: $\mathbf{x}(t+\Delta t) = 2\mathbf{x}(t) - \mathbf{x}(t-\Delta t) + \ddot{\mathbf{x}}(\Delta t)^2$, this method is equivalent to training from trajectory/positions.

### G.4 Hyper-parameters

The default hyper-parameters used for training each architecture is provided below.

•GNODE, CGNODE

| Parameter | Value |
|---|---|
| Node embedding dimension | 5 |
| Edge embedding dimension | 5 |
| Hidden layer neurons (MLP) | 5 |
| Number of hidden layers (MLP) | 2 |
| Activation function | squareplus |
| Number of layers of message passing | 1 |
| Optimizer | ADAM |
| Learning rate | $1.0e^{-3}$ |
| Batch size | 100 |

•**LGN, CLGN, HGN, FGNODE, CFGNODE, FGNN**

| Parameter | Value |
|---|---|
| Node embedding dimension | 8 |
| Edge embedding dimension | 8 |
| Hidden layer neurons (MLP) | 16 |
| Number of hidden layers (MLP) | 2 |
| Activation function | squareplus |
| Number of layers of message passing | 1 |
| Optimizer | ADAM |
| Learning rate | $1.0e^{-3}$ |
| Batch size | 100 |

•**LGNN, CLGNN, HGNN, CHGNN**

| Parameter | Value |
|---|---|
| Node embedding dimension | 5 |
| Edge embedding dimension | 5 |
| Hidden layer neurons (MLP) | 5 |
| Number of hidden layers (MLP) | 2 |
| Activation function | squareplus |
| Number of layers of message passing(pendulum) | 2 |
| Number of layers of message passing(spring) | 1 |
| Optimizer | ADAM |
| Learning rate | $1.0e^{-3}$ |
| Batch size | 100 |

## G.5 Hyper-parameter Search

• **Learning rate**

FGNN

| LR Value | Geometric mean of Zerr | Time (in sec) |
|---|---|---|
| 0.001 | 0.0852 | 4226 |
| 0.003 | 0.0616 | 4394 |
| 0.01 | 0.1219 | 4376 |
| 0.03 | 0.0962 | 4436 |
| 0.1 | 0.0846 | 4476 |
| 0.3 | 0.0628 | 4557 |

GNODE

| LR Value | Geometric mean of Zerr | Time (in sec) |
|---|---|---|
| 0.001 | 0.1371 | 5189 |
| 0.003 | 0.1507 | 5242 |
| 0.01 | 0.1288 | 5250 |
| 0.03 | 0.1267 | 5176 |
| 0.1 | 0.1284 | 5206 |
| 0.3 | 0.1491 | 5179 |

HGNN

| LR Value | Geometric mean of Zerr | Time (in sec) |
|---|---|---|
| 0.001 | 0.1868 | 3856 |
| 0.003 | 0.1895 | 3613 |
| 0.01 | 0.1622 | 3878 |
| 0.03 | 0.1878 | 4089 |
| 0.1 | 0.146 | 9195 |
| 0.3 | 0.2356 | 6859 |

LGNN

| LR Value | Geometric mean of Zerr | Time (in sec) |
|---|---|---|
| 0.001 | 0.1939 | 38082 |
| 0.003 | 0.189 | 11904 |
| 0.01 | 0.1869 | 11890 |
| 0.03 | 0.1879 | 12314 |
| 0.1 | 0.2316 | 11845 |
| 0.3 | 0.279 | 12320 |

• **Number of message-passing layers**

FGNN

| No of message passing layers | Geometric mean of Zerr | Time (in sec) |
|---|---|---|
| 1 | 0.0852 | 6234 |
| 2 | 1.0 | 7552 |
| 3 | 0.1724 | 8024 |
| 4 | 0.0786 | 8455 |

GNODE

| No of message passing layers | Geometric mean of Zerr | Time (in sec) |
|---|---|---|
| 1 | 0.1491 | 7833 |
| 2 | 0.1649 | 6850 |
| 3 | 0.1721 | 6924 |
| 4 | 0.2078 | 7429 |

HGNN

| No of message passing layers | Geometric mean of Zerr | Time (in sec) |
|---|---|---|
| 1 | 0.1782 | 8924 |
| 2 | 0.1796 | 8254 |
| 3 | 0.1583 | 9095 |
| 4 | 0.1608 | 11199 |

LGNN

| No of message passing layers | Geometric mean of Zerr | Time (in sec) |
|---|---|---|
| 1 | 0.1887 | 11649 |
| 2 | 0.2319 | 12345 |
| 3 | 0.1903 | 12172 |
| 4 | 0.1839 | 12838 |

• **Number of hidden layers in MLP**

FGNN

| No of hidden layers | Geometric mean of Zerr | Time (in sec) |
|---|---|---|
| 5 | 0.1132 | 5423 |
| 10 | 0.0848 | 5926 |
| 15 | 0.0821 | 6583 |
| 25 | 0.0934 | 7446 |

GNODE

| No of hidden layers | Geometric mean of Zerr | Time (in sec) |
|---|---|---|
| 5 | 0.2078 | 25559 |
| 10 | 0.1697 | 9246 |
| 15 | 0.2084 | 9728 |
| 25 | 0.185 | 10747 |

HGNN

| No of hidden layers | Geometric mean of Zerr | Time (in sec) |
|---|---|---|
| 5 | 0.1782 | 8984 |
| 10 | 0.1524 | 11307 |
| 15 | 0.1543 | 12605 |
| 25 | 0.1497 | 17498 |

LGNN

| No of hidden layers | Geometric mean of Zerr | Time (in sec) |
|---|---|---|
| 5 | 0.1887 | 12885 |
| 10 | 0.18 | 14954 |
| 15 | 0.1692 | 16533 |
| 25 | 0.1731 | 21011 |

- **Embedding dimensionality in hidden layers of MLP**

FGNN

| No of Neurons | Geometric mean of Zerr | Time (in sec) |
|---|---|---|
| 1 | 0.0646 | 3973 |
| 2 | 0.1132 | 5462 |
| 4 | 0.0763 | 8258 |
| 8 | 0.075 | 13900 |

GNODE

| No of Neurons | Geometric mean of Zerr | Time (in sec) |
|---|---|---|
| 1 | 0.187 | 7664 |
| 2 | 0.2078 | 9895 |
| 4 | 0.1869 | 12258 |
| 8 | 0.7755 | 20041 |

HGNN

| No of Neurons | Geometric mean of Zerr | Time (in sec) |
|---|---|---|
| 1 | 0.1886 | 7689 |
| 2 | 0.1782 | 8984 |
| 4 | 0.1555 | 12615 |
| 8 | 0.1553 | 19389 |

LGNN

| No of Neurons | Geometric mean of Zerr | Time (in sec) |
|---|---|---|
| 1 | 0.2351 | 10448 |
| 2 | 0.1887 | 13070 |
| 4 | 0.18 | 15710 |
| 8 | 0.1674 | 24039 |

- **Activation function**

FGNN

| Activation Function | Geometric mean of Zerr | Time (in sec) |
|---|---|---|
| softplus | 0.886 | 6141 |
| squareplus | 0.1757 | 1340 |

GNODE

| Activation Function | Geometric mean of Zerr | Time (in sec) |
|---|---|---|
| softplus | 0.1287 | 6970 |
| squareplus | 0.1371 | 6425 |

HGNN

| Activation Function | Geometric mean of Zerr | Time (in sec) |
|---|---|---|
| softplus | 0.1953 | 9685 |
| squareplus | 0.1782 | 8463 |

LGNN

| Activation Function | Geometric mean of Zerr | Time (in sec) |
|---|---|---|
| softplus | 0.1921 | 11620 |
| squareplus | 0.1887 | 38082 |

## H   Training and Simulation Time

The key insight obtained from Tables 1-3 is that the LGN family of architectures take significantly longer to train. The GNODE family is marginally faster on average than the HGN family. The LGN family is the slowest since the lagrangian needs to be differentiated, which leads to the differentiation over the GNN parameters. In HGN, the GNN outputs Hamiltonian and thus there is only one layer of differentiation to learn the GNN parameters. Finally, in GNODE, the output is only integrated. Thus, to summarize, in LGN family, the order of differentiation is double, in HGN the order is single and in GNODE, the order is zero.

| Models | Training time (in sec) | Forward Simulation time (in sec) |
|---|---|---|
| CFGNODE | 9475 | 2.21 |
| CGNODE | 8784 | 1.57 |
| CLGN | 55810 | 16.97 |
| CLGNN | 27614 | 3.85 |
| CHGN | 6130 | 0.54 |
| CHGNN | 11038 | 0.82 |
| FGNN | 1325 | 0.02 |
| FGNODE | 8097 | 1.75 |
| GNODE | 6341 | 1.02 |
| LGN | 55042 | 14.82 |
| LGNN | 37752 | 13.53 |
| HGN | 2512 | 0.76 |
| HGNN | 8365 | 0.51 |

Table 1: Training and inference times in pendulum systems.

| Models | Training time (in sec) | Forward Simulation time (in sec) |
|--------|------------------------|----------------------------------|
| FGNN   | 1738                   | 0.01                             |
| FGNODE | 1337                   | 0.25                             |
| GNODE  | 6977                   | 0.10                             |
| LGN    | 141962                 | 5.81                             |
| LGNN   | 25710                  | 0.81                             |
| HGN    | 14053                  | 0.43                             |
| HGNN   | 18128                  | 0.89                             |

Table 2: Training and inference times in spring systems.

| Models | Training time (in sec) | Forward Simulation time (in sec) |
|--------|------------------------|----------------------------------|
| FGNN   | 13136                  | 0.30                             |
| FGNODE | 13238                  | 12.94                            |
| GNODE  | 11422                  | 8.58                             |
| HGN    | 56996                  | 5.21                             |
| HGNN   | 46038                  | 8.04                             |

Table 3: Training and inference times in rigid body systems.