# OpenReview forum: "Unravelling the Performance of Physics-informed Graph Neural Networks for Dynamical Systems"
_NeurIPS.cc/2022/Track/Datasets_and_Benchmarks — NeurIPS 2022 Datasets and Benchmarks _

### Official Review · Reviewer_ETWL · 2022-07-23
**An examination of physics-informed graph neural networks and the strength of their inductive biases**

**Rating:** 7
**Confidence:** 3

**Strengths:**

This reviewer posits that the authors' contributions can be summarized with the following points.
1. The authors systematically analyze the performance of graph neural networks trained and evaluated on different physical systems.
2. The authors demonstrate that enforcing constraints improves the performance of all graph neural network types studied in this work.
3. The authors show that decoupling kinetic and potential energies also improves the performance of all graph neural networks studied.
4. The authors demonstrate that all graph neural networks investigated in this work can generalize to larger systems than the ones on which they were trained.
5. The authors provide readers with an accessible means by which to introduce themselves to the field of deep learning-driven physical modeling.

**Weaknesses:**

This reviewer's perceived limitations of this work are as follows.
1. The authors only conduct experiments on pendulum and spring systems. Rigid body systems, in all their complexities, are not considered in this work.
2. All constraints enforced in this work are specified, not learned.
3. The authors consider only standard variations of a graph neural network architecture. For example, transformer models and their graph locality-focused variants are not considered in this work, even though they have been shown to be equivalent in operations to graph neural networks operating on fully-connected graphs (Reference: https://graphdeeplearning.github.io/files/transformers-are-gnns-slides.pdf).
4. Contacts and deformations between entities in multi-body systems are not considered in this work.

**Additional Feedback:**

This reviewer found the authors' discussion of the utility of graph neural networks for approximating physical quantities such as node-wise acceleration quite insightful. For this reviewer, this raises the question, "In the authors' opinion, are there physical quantities for which graph neural networks would not be optimal to use for approximation?".

**Clarity:**

The paper is fairly well written. However, this work contains a handful of redundant descriptions that should be removed. For example, Section 4.1 (Simulation Environment) repeats its first two sentences immediately thereafter.

**Correctness:**

The authors' claims throughout this work are correct. For example, their benchmarks make use of sound evaluation metrics and report averaged results across several experiments with different initial conditions.

**Documentation:**

The authors have provided a URL to the source code for their data generation and benchmarks. They have also included additional details in their supplementary material to support the reproducibility of their experiments, such as a datasheet and a list of hyperparameters they used throughout their experiments. If it is not already available on GitHub, this reviewer would encourage the authors to also release their source code there to broaden the visibility and accessibility of their work.

**Ethics:**

The authors do not explicitly discuss any ethical concerns that may arise from their work. This reviewer would encourage the authors to consider including a paragraph in their supplementary materials describing the broader impacts of their work and the positive and negative societal effects that such work can have.

**Relation To Prior Work:**

The authors discuss previous works on physics-informed neural networks and inductive graph neural networks. Further, the authors distinguish their work from that of earlier works by summarizing their contributions in terms of their joint investigation of (1) topology-aware modeling; (2) explicit constraints; (3) decoupling of potential and kinetic energies; and (4) the zero-shot generalizability of graph neural networks for physical systems of varying sizes.

**Summary And Contributions:**

In this work, the authors rigorously demonstrate the utility of employing physics-informed graph neural networks for modeling complex dynamical systems. They do so by introducing readers to three main methodological approaches for representing system dynamics and investigating the behavior of these approaches in different physical settings and under physical constraints. Through their benchmarks, the authors present graph neural networks as a foundational architecture for modeling system dynamics, primarily for their inductive biases and subsequent generalizability. Notably, in an accessible manner, this work introduces readers to the field of physical modeling driven by deep learning.

---

> ### Author Response · Authors · 2022-08-25
> **Response to Reviewer ETWL**
>
> We thank the reviewer for the positive comments. Please find the detailed point-by-point response to the comments below.
>
> **The authors only conduct experiments on pendulum and spring systems. Rigid body systems, in all their complexities, are not considered in this work.**
>
> **Response:** We have now included two additional examples, namely, elastic deformation of rigid body and $n$-body gravitational systems to our benchmarking study. These results are now included in the main manuscript (Sec 4.5).
>
> **All constraints enforced in this work are specified, not learned. Contacts and deformations between entities in multi-body systems are not considered in this work.**
>
> **Response:** Since this is a benchmarking study (and accordingly submitted to Benchmarking and Datasets track), we study and evaluate thirteen GNNs available in the literature with the main focus being on physics-informed GNNs. To the best of authors' knowledge, there are no works related to the physics-informed GNNs, where the constraints have been learned. Accordingly, we explicitly specify the constraints. We agree that learning the constraints directly instead of specifying it can be an interesting area of future research. To emphasize this aspect, we now explicitly mention this in the "Concluding Insights" section.
>
> **This reviewer found the authors' discussion of the utility of graph neural networks for approximating physical quantities such as node-wise acceleration quite insightful. For this reviewer, this raises the question, "In the authors' opinion, are there physical quantities for which graph neural networks would not be optimal to use for approximation?"**
>
> **Response:** The utility of GNNs comes to the forefront if there are multiple entities (particles, joints, etc.) interacting with each other. MLPs do not perform message-passing among the interacting entities and hence lack the ability to model the effect of one entity on the other. Hence, for any interacting system GNNs are likely to be superior. MLPs, on the other hand, would be recommended for non-interacting systems such as non-deformable rigid bodies (ball, spinning top, etc.).

---

> > ### Comment · Reviewer_ETWL · 2022-08-29
> > **Helpful Additions**
> >
> > I believe the authors' addition of experiments on rigid body and $\textit{n}$-body gravitational systems have improved the paper's benchmarking scope and, as such, thank the authors for making such additions. To further improve this paper's intended focus area (i.e., benchmarking), I would recommend the authors continue to improve their descriptions of how they intend for future users of these benchmarks to easily be able to develop and benchmark their own methods using the provided source code.
> >
> > Two questions I would like to leave for the authors to think about are as follows. (1) Are users, for example, required to develop future methods in JAX, or will they be able to use other popular machine learning frameworks such as PyTorch? (2) Is the data required to reproduce the authors' benchmarks structured and stored in an accessible manner such that others may easily use the data in other projects?
> >
> > Otherwise, I would like to thank the authors once again for their work in expanding the scope of their benchmarks.

---

> > > ### Author Response · Authors · 2022-08-29
> > > **Thank you and further response**
> > >
> > > We thank the reviewer for the positive comments and insightful suggestions. We shall improve the descriptions in the updated version of the paper. In addition, please find the responses to the points raised.
> > > > (1) Are users, for example, required to develop future methods in JAX, or will they be able to use other popular machine learning frameworks such as PyTorch?
> > >
> > > The codes used in the present work are in the JAX framework primarily due to the differentiable nature of JAX that allows easy modeling of Lagrangian and Hamiltonian neural networks. However, these models can be implemented in other packages such as PyTorch as well.
> > >
> > > >(2) Is the data required to reproduce the authors' benchmarks structured and stored in an accessible manner such that others may easily use the data in other projects?
> > >
> > > We have shared both the data (see: https://doi.org/10.5281/zenodo.7015041) with versioning and the code to generate the data along with the manuscript. In addition, we shall make sure that the data is is structured and stored in an easily accessible fashion in the Zenodo platform with proper documentation so that it can be directly used by future studies.
> > >
> > > Thank you!

---

### Official Review · Reviewer_SgbG · 2022-07-26
**The main observations in the experimental evaluation are not presented clearly.**

**Rating:** 7
**Confidence:** 2
**Correctness:** The claims of the paper sound correct.

**Strengths:**

1- The presentation of the problem, in general, is clear and well-motivated.

2- The authors include three main models along with considering constraints.

2- Including the preliminaries make the paper easier to follow.

3- The limitations and outlook paragraph in the conclusion is appreciated.

**Weaknesses:**


1- The main observations of the experimental evaluation section are confusing and not presented properly. The section starts with mentioning that FGNN is inferior to all other models. Based on what results this conclusion is made? After that, it is mentioned that, like other works, LGN and CLGN were not trained for pendulum systems. Then, FGNN, LGN, and CLGAN are omitted in section 4 and then included in the Appendix. A justification to why these are not trainable is needed in the paper. When the results are included in the Appendix, there is no mention on how these models have become trainable.


2- Since this is an evaluation paper for the goal of benchmarking already existing models that utilize neural networks in the form of MLPs, the impact of hyper-parameters such as number of layers, size of each layer, learning rates, and the type of activation functions (if used) need to be taken into account or at least discussed.



**Additional Feedback:**

Minor comments and suggested improvements:

1- The paragraph between lines 61 and 66 is a repetition of the contributions list.

2- D is not defined in section 2.1.

3- In line 109, it should be "from L", not "from the L".

4- In Section 4.1, the sentence " All the simulations and training were carried out in
184 the JAX environment [22]." is repeated.

5- A missing bracket in the denominator of the EE(t) equation in line 225.

**Clarity:**

The paper is, in general, clear. However, the observations and conclusions drawn from section 4 are not clear.

**Documentation:**

Yes.

**Relation To Prior Work:**

Yes.

**Summary And Contributions:**

This paper evaluates different graph neural networks based systems for modeling dynamical systems.

---

> ### Author Response · Authors · 2022-08-25
> **Response to Reviewer SgbG**
>
> We thank the reviewer for the critical comments. We have addressed all the comments in detail, which has significantly improved the manuscript. Please find the detailed point-by-point response to the comments below.
>
> **Q1- The main observations of the experimental evaluation section are confusing and not presented properly. The section starts with mentioning that FGNN is inferior to all other models. Based on what results this conclusion is made? After that, it is mentioned that, like other works, LGN and CLGN were not trained for pendulum systems. Then, FGNN, LGN, and CLGAN are omitted in section 4 and then included in the Appendix. A justification to why these are not trainable is needed in the paper. When the results are included in the Appendix, there is no mention on how these models have become trainable.**
>
> **Response:** We thank the reviewer for raising these crucial points. The points are addressed one-by-one below.
>
> 1. **FGNN being inferior:** This comment was made by observing the empirical results. We had moved FGNN to the appendix since it was not competitive with respect to the other modes. We realize that this structuring of our manuscript created a gap in the understanding of readers. This has been rectified. Specifically, we have now moved the results of all the models to the main manuscript. It is very clear from the results that FGNN provides significantly inferior performance (See Figs 4,5,8,9). Note that FGNN is a purely data-driven GNN, which learns the dynamics directly from the data with no physics-based inductive biases. Hence, this inferior performance is not surprising.
>
> 2. **LGN and CLGN on Pendulum systems:** Previous work on physics-informed GNNs have focussed only on spring systems and not on pendulum (see: Sanchez-Gonzalez, A., Bapst, V., Cranmer, K. and Battaglia, P., 2019. Hamiltonian graph networks with ode integrators. arXiv preprint arXiv:1909.12790.). Specifically, in pendulum systems, the potential energy of the system depends on the position of the bob. However, in LGN or HGN, the actual position of the particle is not given as an input to the graph, rather the edge distance is given as an input. Thus, it is not possible for LGN or HGN to learn the dynamics of the pendulum system. To address this, we attempted to train LGN and HGN by giving the position of the bob explicitly as a node input feature. However, despite providing this input, we observed that the final model obtained after training using the SOTA practices had high loss in comparison to other other models. Thus, due to the poor performance of the model, we did not include the results of these models in the main manuscript. However, we have now moved all the results from the Appendix to the main manuscript for easy comparison and analysis (See Figs 4-9). It is clear from the updated plots LGN, HGN, CLGN and CHGN gives extremely poor performance in comparison to CLGNN and CHGNN.
>
> We have now included the key insights from the study in a point-by-point fashion in Sec 5. Concluding Insights. In addition to these points, we have also now included additional discussion on training and simulation times of each of the models in the App. H. This clarifies the computational efficiency of each of the models for training and inference.
>
> **Q2- Since this is an evaluation paper for the goal of benchmarking already existing models that utilize neural networks in the form of MLPs, the impact of hyper-parameters such as number of layers, size of each layer, learning rates, and the type of activation functions (if used) need to be taken into account or at least discussed.**
>
> **Response:** We thank the reviewer for raising this point. Detailed hyperparametric search was performed to obtain the optimal values. Specifically, the learning rate, number of message passing layers, number of hidden layers in the MLP, the number of hidden layer units (neurons), and the activation functions were all varied to identify the optimal parameters. These results are now included in the appendix of the updated manuscript (App. G.5).
>
> In addition, to analyze the effect of noise, we have also trained all the models on dataset with Gaussian noise. The results of these models are also included in the updated manuscript (Sec 4.4).
>
> **Minor comments and suggested improvements:**
>
> **1- The paragraph between lines 61 and 66 is a repetition of the contributions list.**
>
> **2- D is not defined in section 2.1.**
>
> **3- In line 109, it should be "from L", not "from the L".**
>
> **4- In Section 4.1, the sentence " All the simulations and training were carried out in 184 the JAX environment [22]." is repeated.**
>
> **5- A missing bracket in the denominator of the EE(t) equation in line 225.**
>
> **Response:** We thank the reviewer for the careful reading. We have now corrected the text.

---

> ### Author Response · Authors · 2022-08-29
> **Looking forward to feedback from Reviewer SgbG**
>
> Dear Reviewer SgbG,
>
> We thank you for taking the time to provide critical comments, which have significantly improved the quality of the manuscript. Since today is the **last day** for author-reviewer discussions, we hope to engage in a discussion and improve the paper to the best extent possible. Specifically, the major changes made in response to the comments by the Reviewer are outlined below.
>
> 1. The results of all the **thirteen models** considered in the study are now included and discussed in the main manuscript. All the relevant figures in the manuscript has been modified to reflect this change.
> 2. Detailed analysis on **hyperparameters** and their effect on the model performance is now included in the manuscript. Tables are provided to show the effect of all the hyperparameters such as learning rate, number of layers, number of neurons in a layer, activation function etc.
> 3. A point-by-point discussion on **important insights** from the study is included in the Conclusions section
>
> With these additional experiments and improved explanations, we believe the Reviewer would now find the manuscript acceptable. Please let us know if there are any additional comments. We would be glad to address those.
>
> We are looking forward to your feedback.
>
> Thank you,
>
> Authors

---

> ### Author Response · Authors · 2022-08-29
> **Eagerly waiting for feedback from Reviewer SgbG**
>
> Dear Reviewer,
>
> Since only a few hours are left till the discussion period closes, we are eager to hear your feedback on our revised version.
>
> We have incorporated the suggestions made by you as well as other reviewers and are happy to inform that two of the reviewers have increased their ratings. With the new additions in our revised submission, we are hopeful that the concerns raised on presentation issues and hyper-parameter tuning have been addressed. We look forward to your feedback.
>
> regards,
>
> Authors

---

### Official Review · Reviewer_iHoy · 2022-07-27
**Comparison of Physics-informed GNNs for spring and pendulum systems**

**Rating:** 5
**Confidence:** 2

**Strengths:**

- A comparison of various physics-inspired GNNs is interesting.
- I like the evaluation metrics used in the paper.
- Efficient implementations in JAX are available.

**Weaknesses:**

- The scope of the paper is very narrow. It looks at pendulum and spring systems. What about other types of systems that GNNs have been used for? Also, the models investigated represent a small subset of all existing surrogate models / learning from dynamical systems models.
- The paper is neither a benchmark paper (it doesn't show how to use the benchmark, or how new methods could be evaluated using the benchmark) nor is it a dataset paper. It doesn't provide enough information on meta-data, data format, API access, maintenance, versioning, etc. of the data.
- Access to the data is through an anonymous repo. Since this is the benchmark and dataset track, I would have expected a repo with DOI, maintenance plan, versioning, etc.
- There seems to be crucial information missing from the main paper. For instance, what specific message-passing GNN was used as the base model?
- One weakness of ML-based models is that the values of hyperparameters matter a lot. For instance, in recent work on GNNs for dynamical systems, it was shown that adding noise to the input at each iteration of really important. The paper doesn't discuss and describe enough the choices made here and the search performed for the hyperparameters used. The appendix lists some of these, but how are they chosen? Randomly?
- The discussion/summary of the findings is cut short and disappoints. What have we now learned from these results? what matters? To have more space one could move a lot of the formulas to the appendix.

**Additional Feedback:**

I am an advocate of taking a step back and benchmarking a large number of existing methods to truly understand "what's going on". However, the submission misses the mark here in execution due to its narrow scope, the suboptimal presentation, and the missing rigor in presenting the setup and results of the experiments.

**Clarity:**

The clarity of the paper could be improved. Details are missing (see above) and for non-expert readers, it might be really difficult to understand how GNNs and the various equations presented at the beginning of the paper are combined. The intuition of how physics-informed GNNs work is missing. Instead, a lot of space is "wasted" on the equations which could be moved to the appendix.

**Correctness:**

There are some typos in the manuscript (I later saw in the appendix it was corrected). other than that, the paper seems correct. I have not checked the implementations though.

**Documentation:**

Essentially, there is no documentation and no information on availability and maintenance, hosting, licensing and maintenance plan

**Ethics:**

Doesn't apply.

**Relation To Prior Work:**

Related work is sufficiently discussed. What isn't clear to this reviewer: why not use existing dynamical systems datasets such as those used in prior work on physics-informed GNNs? Why "only" a comparison on spring and pendulum systems?

**Summary And Contributions:**

The authors compare several existing variations of physics-informed GNNs on the problem of learning dynamical systems in a data-driven way.

---

> ### Author Response · Authors · 2022-08-25
> **Response to Reviewer iHoy**
>
> We thank the reviewer for the critical comments. We have addressed all the comments in detail, which has significantly improved the manuscript. Please find the detailed point-by-point response to the comments below.
>
> **Q1. The scope of the paper is very narrow. It looks at pendulum and spring systems. What about other types of systems that GNNs have been used for? Also, the models investigated represent a small subset of all existing surrogate models / learning from dynamical systems models.**
>
> **Response:** We thank the reviewer for the suggestions. It is worth noting that most of the studies on GNNs for dynamical systems use a purely data-driven approach, where the GNNs are used to learn the position and velocity update directly from the data. Here, we demonstrate that such approaches exhibit poor performance in inferring the dynamics of even simple systems such as $n-$spring and pendulum systems. In addition, even among physics-informed GNNs, we show that the inductive bias in learning provided Lagrangian, Hamiltonian and Newtonian mechanics is differnent, although they in principle are equivalent formulations. Thus, this is the first work, where a wide range of physics-informed GNNs have been carefully analyzed and their performance benchmarked. To address this point, new text has been added to the introduction (Sec 1.) and contributions of the main manuscript.
>
> To make the study more extensive, we have now included **two additional systems**, namely, *gravitational bodies* and an *elastic deformation of a solid*. We evaluate the performance of all the models on these two additional systems. These results are now included in the main manuscript (See Sec 4.5).
>
>
> **Q2. The paper is neither a benchmark paper (it doesn't show how to use the benchmark, or how new methods could be evaluated using the benchmark) nor is it a dataset paper. It doesn't provide enough information on meta-data, data format, API access, maintenance, versioning, etc. of the data. Access to the data is through an anonymous repo. Since this is the benchmark and dataset track, I would have expected a repo with DOI, maintenance plan, versioning, etc.**
>
> **Response:** As mentioned earlier, to the best of authors' knowledge, this is the first attempt to benchmark the performance of a wide range of physics-informed GNNs for learning and inferring the dynamics of physical systems. The code-base of all methods have been made publicly available so that the benchmark can be reproduced, or customized for other scenarios. In addition, we have now provided all the data used for training the models with a DOI in Zenodo at https://doi.org/10.5281/zenodo.7015041. Finally, a key contribution of our work lies in distilling the data obtained from the benchmarking results and transforming them into actionable insights. These insights would enable us to better streamline our research pursuits in the area of physics-informed neural networks, which is currently an active area of research. A summary of the important insights (See Section 5) are as follows:
>
> * Physics-informed GNNs are better than purely data-driven GNNs.
>
> * The inductive bias in learning and inference on GNNs provided by Lagrangian, Hamiltonian, and Newtonian mechanics is different although the formulations are equivalent, in principle.
>
> * Incorporating constraints as inductive biases help.
>
> * Decoupling kinetic and potential energy leads to improved performance. In addition, learning the potential energy at the edge level naturally leads to a stronger enforcement of momentum conservation.
>
> * GNNs allows us to learn from small systems and infer on larger systems with high accuracy. In other words, GNNs facilitate good generalizability.
>
> **Q3. There seems to be crucial information missing from the main paper. For instance, what specific message-passing GNN was used as the base model?**
>
> **Response:** The base model used in the present manuscript is the *full graph network*, which has been widely used to simulate several dynamical systems [1]. Although this has already been mentioned in the manuscript (see Fig. 2 and the very first bullet in Section 3), we have further clarified this in the updated version. In addition, the details of GNODE, LGNN and HGNN (architecture, message passing, and other details and equations) are provided in App. C, D, and E, respectively.
>
> [1] Battaglia, P.W., Hamrick, J.B., Bapst, V., Sanchez-Gonzalez, A., Zambaldi, V., Malinowski, M., Tacchetti, A., Raposo, D., Santoro, A., Faulkner, R. and Gulcehre, C., 2018. Relational inductive biases, deep learning, and graph networks. arXiv preprint arXiv:1806.01261.

---

> > ### Author Response · Authors · 2022-08-25
> > **Part 2**
> >
> > **Q4. One weakness of ML-based models is that the values of hyperparameters matter a lot. For instance, in recent work on GNNs for dynamical systems, it was shown that adding noise to the input at each iteration of really important. The paper doesn't discuss and describe enough the choices made here and the search performed for the hyperparameters used. The appendix lists some of these, but how are they chosen? Randomly?**
> >
> > **Response:** We thank the reviewer for raising this point. Detailed hyperparametric search was performed to obtain the optimal values. Specifically, the learning rate, number of message passing layers, number of hidden layers in the MLP, the number of hidden layer units (neurons), and the activation functions were all varied to identify the optimal parameters. These results are now included in the appendix of the updated manuscript (App. G.5).
> >
> > In addition, to analyze the effect of noise, we have also trained all the models on dataset with Gaussian noise. The results of these models are also included in the updated manuscript (Sec 4.4).
> >
> > **Q5. The discussion/summary of the findings is cut short and disappoints. What have we now learned from these results? what matters? To have more space one could move a lot of the formulas to the appendix.**
> >
> > **Response:** In Q2 above, we summarized the key insights derived from our work. In Section 5 titled "Concluding Insights" we now discuss the insights derived in detail.
> >
> > **Q6. Related work is sufficiently discussed. What isn't clear to this reviewer: why not use existing dynamical systems datasets such as those used in prior work on physics-informed GNNs? Why "only" a comparison on spring and pendulum systems?**
> >
> > **Response:** To the best of the authors' knowledge **physics-informed GNNs** for learning dynamical systems has been analyzed only on spring systems, among the family of particle-based systems (not even pendulum systems, see: Sanchez-Gonzalez, A., Bapst, V., Cranmer, K. and Battaglia, P., 2019. Hamiltonian graph networks with ode integrators. arXiv preprint arXiv:1909.12790.). Indeed, more complex systems have been studied using purely data-driven GNNs. To address this comment, we have now included two additional systems, namely, gravitational bodies and elastically deformable 3D solid body. The performance of all the models on these systems are now evaluated. These additional results are now included in the main manuscript in Sec 4.5.
> >
> > **Q7. The clarity of the paper could be improved. Details are missing (see above) and for non-expert readers, it might be really difficult to understand how GNNs and the various equations presented at the beginning of the paper are combined. The intuition of how physics-informed GNNs work is missing. Instead, a lot of space is "wasted" on the equations which could be moved to the appendix.**
> >
> > **Response:** Most of the work on GNNs for physical systems employ a graph architecture to model the system and learn the dynamics directly from the data. However, we show that by infusing physics-based inductive biases, the performance of GNNs can be significantly enhanced. Specifically, instead of learning directly from the data, physics-informed GNNs take the position and velocity as input and predict abstract quantities such as energy of force as an output. These output values are then used along with the physics-based equations to obtain the updated trajectory. It should be noted that the loss function to train the GNN models are on the trajectory in all the cases. Thus, in physics-informed GNNs, the graph essentially learns the function relating the position and velocity to quantities such as force or energy, by training on trajectory. In addition, in the present work, we carefully analyze the inductive bias provided by Lagrangian, Hamiltonian, and Newtonian mechanics in GNNs for learning and inferring the dynamics. We show that, although these mechanics are equivalent in principle, the inductive bias provided by them is clearly distinct. This is reflected in the performance of each of these models as elucidated in the present work.
> >
> > To address this comment, we have provided additional explanations in the Introduction (Sec. 1), and Concluding insights (Sec. 5). We have also included a new section Sec 2.4, which summarizes how the physics-based inductive biases are combined with the GNN. In addition to these points, we have also now included additional discussion on training and simulation times of each of the models (see Concluding Insights and App. H). This clarifies the computational efficiency of each of the models for training and inference.

---

> > > ### Author Response · Authors · 2022-08-25
> > > **Part 3**
> > >
> > > **Q8. I am an advocate of taking a step back and benchmarking a large number of existing methods to truly understand "what's going on". However, the submission misses the mark here in execution due to its narrow scope, the suboptimal presentation, and the missing rigor in presenting the setup and results of the experiments.**
> > >
> > > **Response:** With the additional results and explanations, we now hope the reviewer finds the manuscript suitable for publication. Please do let us know if there are further queries.

---

> > > > ### Comment · Reviewer_iHoy · 2022-08-29
> > > > **Right direction**
> > > >
> > > > The improvements go in the right direction. Thanks for your response and the additions to the paper.
> > > >
> > > > I’m happy to increase my score slightly but still think the paper has weaknesses that have not or cannot be sufficiently addressed.
> > > >
> > > > There must have been a misunderstanding regarding the noise. My comment was not asking to test robustness to noise. The comment was intended to make you aware of the importance of noise in training graph networks on physical systems data. Noise as a way to improve generalization. Here it’s not enough to simply try one type of noise (standard Gaussian noise) and leave it at that. It has been shown numerous times in prior work (even the worm you refer to as purely data driven) that this can make a huge difference. You need to experiment with different means etc
> > > >
> > > > My criticism of the submission neither being a benchmark nor a dataset paper largely remains. To provide the necessary API, persistent storage, meta-data etc seems more like an afterthought to a submission that looks more like a paper comparing some methods on existing datasets. It’s a nice set of experiments, but the benchmark/dataset perspective is missing. Here I believe it is more important to provide a means for others to test future methods and datasets and lots more thoughts on an API etc and less method/result driven content. Also, I stand by my comment regarding the scope being quite narrow.
> > > >
> > > > Nevertheless, I’ll increase my score slightly.

---

> > > > > ### Author Response · Authors · 2022-08-29
> > > > > **Thank you and further response to Reviewer iHoy**
> > > > >
> > > > > We thank the reviewer for raising the score and providing additional comments.
> > > > > 1. **Regarding the noise**: We understand that the reviewer was suggesting it as a means to improve generalization. This shall be performed and included at a later version of the manuscript.
> > > > > 2. **Benchmark/dataset perspective**: Thank you for raising this comment. As mentioned earlier, the aim of the present manuscript is to benchmark the performance graph neural networks for modeling physical systems---a first of its kind study. Future works on modeling physical systems can thus follow the datasets, metrics and evaluations mentioned in the present work to benchmark their model against the existing GNNs. We shall improve the text to incorporate this perspective in detail.
> > > > >
> > > > > Thank you!

---

> ### Author Response · Authors · 2022-08-29
> **Looking forward to feedback from Reviewer iHoy**
>
> Dear Reviewer iHoy,
>
> We thank you for taking the time to provide critical comments, which have significantly improved the quality of the manuscript. Since today is the **last day** for author-reviewer discussions, we hope to engage in a discussion and improve the paper to the best extent possible. Specifically, the major changes made  in response to the comments by the Reviewer are outlined below.
> 1. Additional experiments on **gravitational and deformable body** systems for all the GNNs considered.
> 2. Access to data through a **repository with DOI**.
> 3. Discussion on the **important metrics** to benchmark a GNN for simulating physical systems such as energy, and momentum conservation and rollout. Specifically, we bring out the importance of **momentum error**, which remain neglected in the literature.
> 4. Detailed analysis on **hyperparameters** and their effect on the model performance.
> 5. A point-by-point discussion on **important insights** from the study in the Conclusions section.
>
> With these additional experiments and improved explanations, we believe the Reviewer would now find the manuscript acceptable. Please let us know if there are any additional comments. We would be glad to address those.
>
> We are looking forward to your feedback.
>
> Thank you,
>
> Authors

---

### Official Review · Reviewer_mtmX · 2022-07-28
**Valuable new benchmark for physics-informed GNNs, with some improvements possible**

**Rating:** 7
**Confidence:** 4

**Strengths:**

1. Careful comparison and evaluation of multiple different physics-informed GNN models plus the impact of external constraints
2. Development of a standardized set of dynamical systems (n-pendulum and n-spring), with extrapolation beyond the training domain
3. Examination/explanation of relative strengths of different GNN models and impact of external constraints

**Weaknesses:**

1. Other dynamical systems (like n-bodies in a gravitational field, etc.) would have been interesting to include
2. No discussion of how to incorporate measurement noise or bias or if it’s at all possible in any of these frameworks


**Additional Feedback:**

Additional feedback is as follows. In general, try to re-run grammar and spell checker.

Content

* L94 Could you clarify why constraints must be of the form $A(x)\dot{x}$ = 0 ?
Eq. (4) Missing closing right parentheses somewhere. Please check and correct.
* L99 from Eq. 2
* L103 In this paper, is an explicit time dependence for T and V or an explicit x dependence for T ever considered, i.e. $T(x, \dot{x}, t)$, $V(x, t)$? If not, can this be simplified to $T(\dot{x})$, $V(x)$ already? Or is the point that the model may not be constrained in this way in some situations?
* Eq. (7) The notation $\nabla_{\dot{x}\dot{x}}$ is confusing. Does this mean a double gradient or d'Alembertian, i.e. is it a vector again or a scalar?
* L113 This is a bit repetitive as it was already explained that $\mathbf{M} = \nabla_{\dot{x}\dot{x}} L$
* L119 Is there a sign error here? I think I get the correct equations if I assume $J = [0, I; -I, 0]$. Please check.
* Eq. (12) Choose one of $D_z$ or $D_Z$ for consistency
* Eq. (13) Correct error as stated in Supplementary material
* L312 Could this also be related to Noether’s theorem?
* Eq. (18) (Supplementary) Is this correct? If I naively differentiate Eq. (17), I get $(x_i - x_{i-1})(\dot{x} _i - \dot{x} _{i-1}) = 0$, which does not seem to reduce to Eq. (18)

Language/style

* L39 (HNNs)
* L40 Langrangian or Hamiltonian
* L52 they can potentially learn
* L61 based on the LNN, HNN, and NODE frameworks
* L159 improvises does not seem like the correct word here
* L183 Two sentences are repeated twice unnecessarily
* L437 (Supplementary) “Pfaffian” should not be in math mode


**Clarity:**

Yes, the paper well written, but can benefit from another pass of grammar and spell check and minor corrections to formulas.

**Correctness:**

Yes, the claims appear to be correct and the dataset is constructed in a sound way. The evaluation methods and experiment design are also appropriate and performed correctly.

**Documentation:**

Dataset is currently hosted as an anonymous GitHub repo, which includes code for how to generate it. I would recommend the datasets (as used in the paper) should also be uploaded to a platform like Zenodo (to generate a persistent DOI) and fully documented in accordance with FAIR principles: https://www.go-fair.org/fair-principles/


**Ethics:**

No ethical concerns.

**Relation To Prior Work:**

Yes, the relation to prior work is discussed. This work differs in that it compares different architectures on a fair footing. However, there can be some additional citations to, e.g. https://arxiv.org/abs/1612.00222

**Summary And Contributions:**

This paper develops a set of benchmarks consisting of datasets and metrics for physics-informed graph neural networks (GNNs) for dynamical systems. In particular, they examine and compare the efficacy of Hamiltonian and Lagrangian GNNs and graph neural ODEs in terms of rollout error, and energy and momentum violations for scenarios including n-pendulum and n-spring systems. They provide the code to generate the datasets, train the models, and evaluate the metrics.

---

> ### Author Response · Authors · 2022-08-25
> **Response to Reviewer mtmX**
>
> We thank the reviewer for the critical comments. We have addressed all the comments in detail, which has significantly improved the manuscript. Please find the detailed point-by-point response to the comments below.
>
> **Q1. Other dynamical systems (like n-bodies in a gravitational field, etc.) would have been interesting to include.**
>
> **Response:** We thank the reviewer for this suggestion. We have now included benchmarking of all the models on two additional systems namely, gravitational bodies and elastic deformation in a 3D solid to make the study more extensive. These results are now included in the main manuscript (Sec 4.5 and Appendix B.3-B.4). A snapshot of the systems studied are added as Figure 3.
>
> **Q2. No discussion of how to incorporate measurement noise or bias or if it’s at all possible in any of these frameworks**
>
> **Response:** We have now trained all the models on data with Gaussian noise. Our experiments reveal that the error increases 10-fold when trained on noisy data (see Sec 4.4). However, the trends in terms of ordering of architectures based on their performance remain the same. This result indicates a future direction of research, which is to build physics-informed GNNs that are more robust to noise.
>
> **Q3. Yes, the relation to prior work is discussed. This work differs in that it compares different architectures on a fair footing. However, there can be some additional citations to, e.g. https://arxiv.org/abs/1612.00222**
>
> **Response:** Thank you for pointing this out. We have now cited the referred work.
>
> **Q4. Dataset is currently hosted as an anonymous GitHub repo, which includes code for how to generate it. I would recommend the datasets (as used in the paper) should also be uploaded to a platform like Zenodo (to generate a persistent DOI) and fully documented in accordance with FAIR principles: https://www.go-fair.org/fair-principles/**
>
> **Response:** We have now shared the training data in Zenodo at https://doi.org/10.5281/zenodo.7015041.
>
> **Q5. L94 Could you clarify why constraints must be of the form $A(x)\dot{x}= 0$? Eq. (4) Missing closing right parentheses somewhere. Please check and correct.**
>
> **Response:** A constraint on a system essentially restricts the motion of a system to a subspace among all the allowable paths. For instance, in the case of two particles with the coordinates $(x,y)$ and $(0,0)$ connected by an inextensible rod, the constraint equation can be given as $(x^2 + y^2) =l^2$. Such constraints are known $holonomic$ constraints. However, a different set of constraints act in cases such as multifingered grasping, known as *Pfaffian* constraints, where instead of positions, constraints are enforced on velocities. The generic form of a *Pfaffian* constraint is $A(x)\dot{x}= 0$. Note that any $holonomic$ constraint can also be written in the form of a *Pfaffian* constraint by differentiating the original form. For instance, the constraint equation for two particles mentioned earlier can be differentiated to obtain $x\dot{x}+y\dot{y}=0$, which is of the form $A(x)\dot{x}= 0$. For the sake of generality, here we adopt this form to express the constraints. More details on this can be found in the Section 1, Chapter 6 of Murray et al. (Murray, R.M., Li, Z. and Sastry, S.S., 2017. A mathematical introduction to robotic manipulation. CRC press.)
>
> To clarify this point, we have added additional text in the main manuscript and reference to the additional reading material (See Section 2.1 and Appendix A).
>
> **Q6. L99 from Eq. 2**
>
> **Response:** Corrected.
>
> **Q7. L103 In this paper, is an explicit time dependence for T and V or an explicit x dependence for T ever considered, i.e., $T(x,\dot{x},t),V(x,t)$? If not, can this be simplified to $T(\dot{x}),V(x)$ already? Or is the point that the model may not be constrained in this way in some situations?**
>
> **Response:** The dependence presented in the paper is the most generic expression of the Lagrangian, that is, $T(x,\dot{x},t),V(x,t)$. Indeed, in the case of particle-based systems, this reduces to $T(\dot{x},t),V(x,t)$. However, in the case of rigid bodies such as articulated bodies, kinetic energy can be a function of the position as well as the velocity. Thus, while describing the Lagrangian mechanics, the generic form is adopted.
>
> **Q8. Eq. (7) The notation $\nabla_{\dot{x}\dot{x}}$ is confusing. Does this mean a double gradient or d'Alembertian, i.e. is it a vector again or a scalar?**
>
> **Response:** The notation $\nabla_{\dot{x}\dot{x}}$ refers to $\frac{\partial^2}{\partial \dot{x}^2}$, which when applied on the Lagrangian results in the mass matrix M. This is now clarified in the text following Eq. 7.
>
> **Q9. L113 This is a bit repetitive as it was already explained that
> $\textbf{M}=\nabla_{\dot{x}\dot{x}}L$**
>
> **Response:** Thank you. The text is now corrected to remove the redundancy.

---

> > ### Author Response · Authors · 2022-08-25
> > **Part 2**
> >
> > **Q10. L119 Is there a sign error here? I think I get the correct equations if I assume $J=[0,I;−I,0]$. Please check.
> > Eq. (12) Choose one of $D_z$ or $D_Z$ for consistency
> > Eq. (13) Correct error as stated in Supplementary material**
> >
> > **Response:** Thank you for the careful reading. All the minor errors have been corrected.
> >
> > **L312 Could this also be related to Noether’s theorem?**
> >
> > **Response:** We thank the reviewer for bringing this point up. Indeed, it is related to Noether's theorem. The translational symmetry in position, which is better enforced due to the decoupling of kinetic and potential energy and computing the potential energy at the edge level, could be the reason why the momentum conservation is better enforced in LGNN and HGNN systems in comparison to other systems. To clarify this, additional text has been added in the Sec. 4.3.2 of the main manuscript.
> >
> > **Eq. (18) (Supplementary) Is this correct? If I naively differentiate Eq. (17), I get $(xi−x{i−1})(\dot{x}i−\dot{x}{i−1})=0$, which does not seem to reduce to Eq. (18)**
> >
> > **Response:** Indeed, thank you for the careful reading. These equations are now corrected in the App. B.1.
> >
> > **Language/style
> > L39 (HNNs)
> > L40 Langrangian or Hamiltonian
> > L52 they can potentially learn
> > L61 based on the LNN, HNN, and NODE frameworks
> > L159 improvises does not seem like the correct word here
> > L183 Two sentences are repeated twice unnecessarily
> > L437 (Supplementary) “Pfaffian” should not be in math mode**
> >
> > **Response:** Thank you for the careful reading. All the minor errors have been corrected.

---

> > > ### Comment · Reviewer_mtmX · 2022-08-28
> > > **Reponses**
> > >
> > > Thank you for your responses. As they address my main concerns, I have raised my score.
> > >
> > > The addition of the robustness to noise, and the addition of the new benchmarks strengthen the work considerably. However, I question the exact choice of Gaussian noise experiments. Is standard deviation of 1 meaningful? Is that a large amount of noise or a small amount of noise? I would think it depends a lot on the range/scale of the measurement in the particular dataset. Instead, I would suggest injecting some relative uncertainty (say 10%) and perhaps even scanning it (20%, 50%, etc.), as that will give a much better indication of which models are more robust to noise (and at what level).

---

> > > > ### Author Response · Authors · 2022-08-29
> > > > **Thank you and further response to Reviewer mtmX**
> > > >
> > > > We thank the reviewer for the positive comments and raising the score.
> > > >
> > > > In order to address the additional concerns raised, we have performed a parametric study with relative uncertainty as noise for spring systems as suggested by the reviewer. Specifically, we analyzed the performance of models for **1%, 5%, 10% and 50% noise** with respect to the standard deviation (see Fig. 21 and 22). We observe that the model performance deteriorates with increasing noise. Please note that due to limited availability of time to train the models, the study is limited to only spring systems. Further, due to the large amount of training time taken by LGN, it was not as well trained as other models. These issues shall be addressed in the later version of the manuscript.
> > > >
> > > > To address this comment, **two new figures (Figs. 21 and 22)** are added to the appendix.
> > > >
> > > > Thank you!

---

### Author Response · Authors · 2022-08-25
**General comments. Applies to all the reviewers**


We thank the reviewers for their insightful comments and suggestions. Please find a point-by-point response to the comments raised by the reviewers below. We have also updated the main manuscript and the appendix to address these comments. The changes made in the main manuscript are highlighted in blue color. The **major changes** made in the manuscript are listed below.
1. We have now included benchmarking of all the models on **two additional systems** namely, gravitational bodies and elastic deformation in 3D solid on which all the models are evaluated and a detailed discussion is included (Sec. 4.5 and Figs. 10, 15, 16). A new figure showing the **visualization of all the experimental systems** studied is included (Fig. 3).
2. **All the data used** for the training has been shared explicitly with versioning in Zenodo at https://doi.org/10.5281/zenodo.7015041
3. We have evaluated and included the performance of all the models trained on data with **Gaussian noise**. A detailed discussion on this is included in the main manuscript (Sec. 4.4 and Figs. 17, 18, 19, and 20).
4. Detailed results on **hyperparametric variation** for each model are included (Sec. G.4 and G.5).
5. All the plots in the main manuscript have been updated to include the results of all the GNNs. Thus a total of **thirteen versions** of GNNs are now evaluated and discussed in the main manuscript (Figs. 4-9).
6. **Key insights** drawn from our study are given in a point-by-point fashion in Sec. 5 Concluding insights.
7. A new section on the **physics-informed GNNs** are added to the main manuscript (Sec. 2.4).

---

### Author Response · Authors · 2022-08-27
**Looking forward to post-rebuttal feedback**

Dear Reviewers,

Thank you once again for all of your constructive comments, which have helped us significantly improve the paper! As detailed below, we have performed several additional experiments and analyses to address the comments and concerns raised by the reviewers.

Since the discussion phase is going to end in two days, we are eagerly looking forward to your post-rebuttal responses.

Please do let us know if there are any additional clarifications or experiments that we can offer. We would love to discuss more if any concern still remains. We appreciate your suggestions.

Thank you!

---

### Meta-Review · Area_Chair_wbPM · 2022-09-15

**Recommendation:** Accept
**Confidence:** 4

**Metareview:**

This paper evaluates the use of physics-informed GNNs for modeling dynamical systems (by simulating them) and explores the properties that are beneficial for achieving enhanced performances with them. The main concerns of the reviewers from the initial version have been addressed during the rebuttal and discussion periods, and at this point most reviewers are in favor of acceptance (with the remaining one only marking a marginal score). Therefore, I also recommend accepting the paper.

---

### Decision · Program_Chairs · 2022-09-16

Accept